# Gene-SGAN: discovering disease subtypes with imaging and genetic signatures via multi-view weakly-supervised deep clustering

Disease heterogeneity has been a critical challenge for precision diagnosis and treatment, especially in neurologic and neuropsychiatric diseases. Many diseases can display multiple distinct brain phenotypes across individuals, potentially reflecting disease subtypes that can be captured using MRI and machine learning methods. However, biological interpretability and treatment relevance are limited if the derived subtypes are not associated with genetic drivers or susceptibility factors. Herein, we describe Gene-SGAN – a multi-view, weakly-supervised deep clustering method – which dissects disease heterogeneity by jointly considering phenotypic and genetic data, thereby conferring genetic correlations to the disease subtypes and associated endophenotypic signatures. We first validate the generalizability, interpretability, and robustness of Gene-SGAN in semi-synthetic experiments. We then demonstrate its application to real multi-site datasets from 28,858 individuals, deriving subtypes of Alzheimer's disease and brain endophenotypes associated with hypertension, from MRI and single nucleotide polymorphism data. Derived brain phenotypes displayed significant differences in neuroanatomical patterns, genetic determinants, biological and clinical biomarkers, indicating potentially distinct underlying neuropathologic processes, genetic drivers, and susceptibility factors. Overall, Gene-SGAN is broadly applicable to disease subtyping and endophenotype discovery, and is herein tested on disease-related, genetically-associated neuroimaging phenotypes.

Neurologic and neuropsychiatric diseases are associated with pathologic processes, which lead to heterogeneous brain changes modified by underlying genetic determinants, as well as lifestyle and environmental factors. Imaging has been a cornerstone of studying the human brain over the past three decades, enabling the observation and measurement of these changes in vivo[1], thereby deepening our understanding of how aging and diseases affect brain structure and function. The combination of artificial intelligence (AI) methods and imaging has recently allowed us to transcend the limitations of patient-control comparisons and identify imaging signatures on an individual basis, thereby deriving imaging-AI (iAI) signatures for early disease detection and individualized prognostication[2–4]. However, many such iAI signatures have been developed independently of underlying genetic influences, despite the increasing evidence for strong associations between these iAI biomarkers and genetic variants in brain diseases[5–8]. This has limited their biological interpretability and ability to provide mechanistic insights, as well as their clinical applicability for potential gene-guided therapy and drug discovery[9].

✉ e-mail: Christos.Davatzikos@pennmedicine.upenn.edu

The heterogeneity of brain diseases and aging poses further challenges in the biological interpretability and clinical utility of these iAI signatures. In particular, multiple co-occurring pathologic processes can simultaneously and jointly affect the brain. For example, amyloid plaques, tau tangles, and medial temporal lobe neurodegeneration are hallmarks of Alzheimer's pathology, whereas cerebrovascular diseases, frequently co-existing with AD, also contribute to cognitive decline and neurodegeneration with distinct but overlapping effects[10]. Various commonly obtained imaging measurements such as volumes of brain structures, cortical thickness, or strength of functional networks lack specificity by virtue of being affected by multiple such pathologies. Methods for disentangling such heterogeneity[11–15] can enable precision diagnostics and disease subtyping by identifying the type and extent of pathologic processes that actively influence an individual's brain phenotype.

To address these challenges, we develop a deep learning method, Gene-SGAN (Gene-guided weakly-supervised clustering via generative adversarial networks), to model the heterogeneity of disease effects by estimating respective endophenotypic iAI signatures that reside inside the causal pathway from genetic variants to disease symptoms/diagnosis, which may be considered an 'exophenotype'[16]. Critically, by linking imaging phenotypes with genetic factors, Gene-SGAN endorses biologically interpretable in vivo measurements of genetically-associated brain changes related to pathologic processes and diseases, or an 'endophenotype'. Based on the expression of endophenotypic iAI signatures, Gene-SGAN clusters patients into disease subtypes with relatively more homogeneous and genetically associated brain phenotypes. This subtyping aims to contribute to precision diagnostics, patient stratification into clinical trials, and a better understanding of heterogeneous neuropathologic processes giving rise to similar clinical symptoms.

The foundation of our methodology is a deep learning architecture that links imaging and genetic data in a latent space encoding genetically-associated imaging subtypes of brain pathologies. Critically important in our approach is the generative modeling of pathologic processes, such as effects of a disease or a risk factor on brain structure, via a GAN[17] which maps brain measurements from a reference population (healthy controls) to a target population (disease cohort). This generative modeling of pathologic brain change is linked to genetic risk factors for a disease or a clinical condition. Moreover, clustering in the latent space directly leads to disease subtyping according to these genetically-associated brain phenotypes. Several mechanisms (Method 1) regularize this process, thereby enabling the stability, robustness, and interpretability of the derived subtypes.

In this work, we present GeneSGAN from the perspective of brain aging and dementia, although it is a general methodology. We first demonstrate its generalizability, interpretability, and robustness through semi-synthetic experiments using data from seven studies. Subsequently, leveraging data from 28,858 individuals from two large studies, we seek to unravel genetically-linked heterogeneity of neuroanatomical changes in two different populations: (1) patients with clinical AD or Mild Cognitive Impairment (MCI); and (2) cognitively normal older adults with hypertension, a known risk factor for cerebrovascular diseases that contribute to dementia[18]. Within both of these populations, Gene-SGAN identifies reproducible subtypes distinguished by their replicable neuroanatomical patterns, genetic underpinnings, and clinical profiles.

## Results
### Gene-SGAN: gene-guided weakly-supervised clustering via generative adversarial network to derive disease-related subtypes with distinctive imaging and genetic signatures
Gene-SGAN aims to identify genetically-associated disease subtypes from phenotypic and genetic features. In the present work, specifically, we focused on brain phenotypic features derived from magnetic

resonance imaging (MRI) and disease-associated single nucleotide polymorphisms (SNPs) as genetic features. The methodological advances of Gene-SGAN are two-fold. First, a deep generative model learns one-to-many mappings from phenotypic measures of a reference population (e.g., brain measurements from healthy controls) to those of a target population (e.g., a patient cohort), thereby capturing the diversity of brain change patterns related to disease. This approach aims to reduce confounders from disease-unrelated variations such as demographic factors or disease-unrelated genetic influences on the brain phenotype. Second, a low-dimensional latent space in Gene-SGAN unravels phenotypic and genetic heterogeneity into latent variables that reflect disease subtypes. In particular, the latent space separately encodes phenotypic subtypes associated with genetic factors through the variables $z_1$, while capturing unlinked phenotypic and genetic variations via two ancillary sets of variables $z_2$ and $z_3$, respectively (Fig. 1a).

For estimating heterogeneous disease effects on brain phenotypic features, one-to-many mappings were constructed via a GAN that learns a transformation function mapping the reference phenotypic features into various types of generated features (Fig. 1b). The latent variables $z_1$ and $z_2$ influence the transformation function and aim to summarize disease-related brain variations among the target populations. As typical in GANs, a discriminator attempts to distinguish the real from the generated disease effects on the brain phenotype, thereby ensuring that the generated brain features follow the distribution of the real target brain features. As a key component of this framework, an inverse mapping is introduced to re-estimate the latent variables $z_1$ and $z_2$ from the generated target features so that the latent variables capture distinct and recognizable brain signatures, which contributes to interpretability of respective iAI phenotypes. Estimation of these unknown latent variables with comparison to a reference population is referred to as weakly-supervised learning in this study.

Gene-SGAN also incorporates genetic features into the model framework to identify disease-related subtypes with genetic underpinnings. Through a Variational Inference (VI) approach (Fig. 1b), the model approximates the distribution of genetic features based on the latent variables $z_1$ and $z_3$ through a decoding neural network. In particular, disease-related phenotypic signatures associated with genetic features are summarized by the variable $z_1$, which is estimated by the same inverse mapping incorporated in the GAN training process. Moreover, another encoding neural network estimates the posterior distribution of the latent variable $z_3$ that captures the genetic variance not reflected by brain characteristics. Refer to Method 1 for mathematical details.

Incorporated in the frameworks for both GAN and VI, the M-dimensional categorical variable $z_1$ is the key latent variable that characterizes disease-related variations induced by phenotypic-genetic associations. The inverse mapping of the trained Gene-SGAN model is applied to participants' phenotypic features only to estimate their latent variables $z_1$, which indicate their probabilities belonging to the $M$ subtypes that display genetic associations. Each participant was subsequently assigned to a single subtype based on the maximum probability.

### Semi-synthetic experiments
We validated the Gene-SGAN model in extensive semi-synthetic experiments, using regions of interest (ROIs) from brain MRI and SNP as input phenotypic and genetic features. Using real HC participant ROI data, we constructed Pseudo-patient (Pseudo-PT) data, which we refer to as semi-synthetic data. Specifically, to construct the ROIs of Pseudo-PT participants, we imposed synthetic volumetric change in a set of predefined ROIs on the real HC participants' ROI measures to simulate disease effects. Simultaneously, we constructed completely synthetic genetic data modulating disease effects by randomly sampling minor allele counts of 100, 250, or 500 simulated SNPs. These

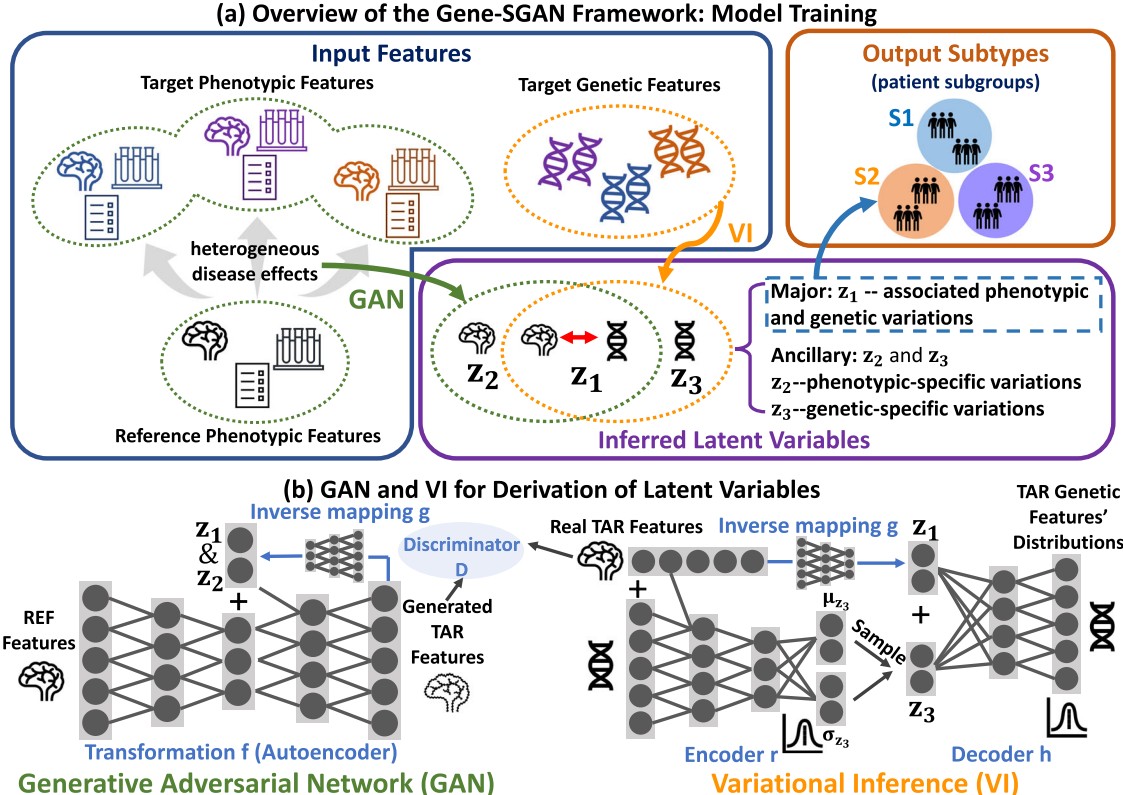

**(a) Overview of the Gene-SGAN Framework: Model Training**

**(b) GAN and VI for Derivation of Latent Variables**

**Fig. 1 | Gene-SGAN identifies disease-related subtypes simultaneously guided by genetic and phenotypic features.** *Subtypes*: patient clusters based on genetically-explained phenotypic variations (e.g., brain neurodegeneration) associated with pathologic processes; *genetic features*: disease-associated genetic factors (e.g., disease-associated SNPs); *phenotypic features*: features from clinical phenotype data, such as imaging features obtained from brain MRI. **a** Overview of the Gene-SGAN framework, which aims to identify disease-related subtypes by deriving latent variables $z_1$ that capture linked phenotypic and genetic variations. To avoid bias in $z_1$, two ancillary latent variables, $z_2$ and $z_3$, are learned to capture phenotype-specific and genetic-specific variations, respectively. Particularly, $z_1$ and $z_2$ are learned through a GAN that models one-to-many mappings from a reference (REF) group's (e.g., health control (HC) population) to a target (TAR) group's (e.g., patient population) phenotypes, so that they capture disease effects on normal phenotypic features rather than variance affected by disease-unrelated factors. A Variational Inference (VI) approach further encourages the genetic associations of $z_1$ and $z_3$. Taken together, through $z_1$, $z_2$, and $z_3$, our approach identifies disease-

related subtypes with associated phenotypic patterns and genetic underpinnings. **b** GAN and VI are trained iteratively to derive the latent variables. First, to model one-to-many mappings from REF to TAR populations, Gene-SGAN utilizes GAN to learn a transformation function f that generates TAR phenotypic features from REF phenotypic features. As inputs of f, the latent variables $z_1$ and $z_2$ control the disease-related variations in the generated TAR features (i.e., mapping directions). An inverse mapping g is introduced to re-estimate $z_1$ and $z_2$ from the generated TAR features, ensuring that the latent variables characterize sufficiently different and recognizable phenotypic variations. Second, the VI approach estimates the posterior distribution of $z_3$ (i.e., mean $\mu_{z_3}$ and std $\sigma_{z_3}$) based on the TAR phenotypic and genetic features through an encoding neural network r. Simultaneously, a decoding neural network h infers the distribution of TAR genetic features conditioned on $z_1$ and sampled $z_3$. Here, $z_1$, estimated by the same inverse mapping g in GAN, summarizes necessary information on the TAR phenotypic features for inferring the TAR genetic features' distributions. The plus sign denotes feature concatenation.

Pseudo-PT participants were divided into three ground truth subtypes. Each subtype shared one similar imaging pattern (Fig. 2e) and had higher minor allele frequencies (MAFs) in four selected SNPs compared to the remaining Pseudo-PT participants (Fig. 2d). Moreover, we introduced confounding imaging patterns to subsets of Pseudo-PT participants who did not have shared genetic features (Fig. 2e). Similarly, higher MAFs in confounding SNPs were simulated to subsets of Pseudo-PT participants without shared imaging patterns. In the simulation, we retained disease-unrelated variations in imaging features and provided the known ground truth of simulated imaging patterns, SNPs, and subtypes. More details of data simulation are presented in Method 6.

**Gene-SGAN is generalizable to test data**
The generalizability of Gene-SGAN's performances to test data underpins the reliability of derived subtypes. We evaluated Gene-SGAN's generalizability and examined hyperparameter selections in the semi-synthetic experiments (Method 6). To this end, we adopted a 50-repetition holdout cross-validation (CV, 20% for testing) procedure. Gene-SGAN consistently achieved comparable clustering accuracies

on the training and test sets, endorsing the robustness of the model's generalizability (Fig. 2a, Supplementary Fig. 1). Furthermore, the hyperparameter (gene-lr) impacted the clustering performance. Gene-lr regulates the importance of genetic features relative to imaging features in optimization (Method 1). With increasing simulated imaging confounders in datasets (from one to two), a higher gene-lr ($4 \times 10^{-4}$ vs. $1 \times 10^{-4}$) led to the optimal performance, suggesting the reliance on genetic data to guide the clustering solution when many non-genetically related confounding factors play a role (Fig. 2a).

The optimal gene-lr in different cases was selected through the CV procedure with a selection metric, N-Asso-SNPs (i.e., the number of significantly associated SNPs in the test set) (Method 6). N-Asso-SNPs was positively associated with the clustering accuracy (Supplementary Fig. 1), thus serving as an appropriate metric for selecting optimal hyperparameters in real data applications.

**Gene-SGAN is robust to missing SNPs**
Missing data are common in genomics using SNPs. Gene-SGAN was designed to handle missing SNPs as a multivariate learning model. To simulate this situation and test the model's robustness, we randomly

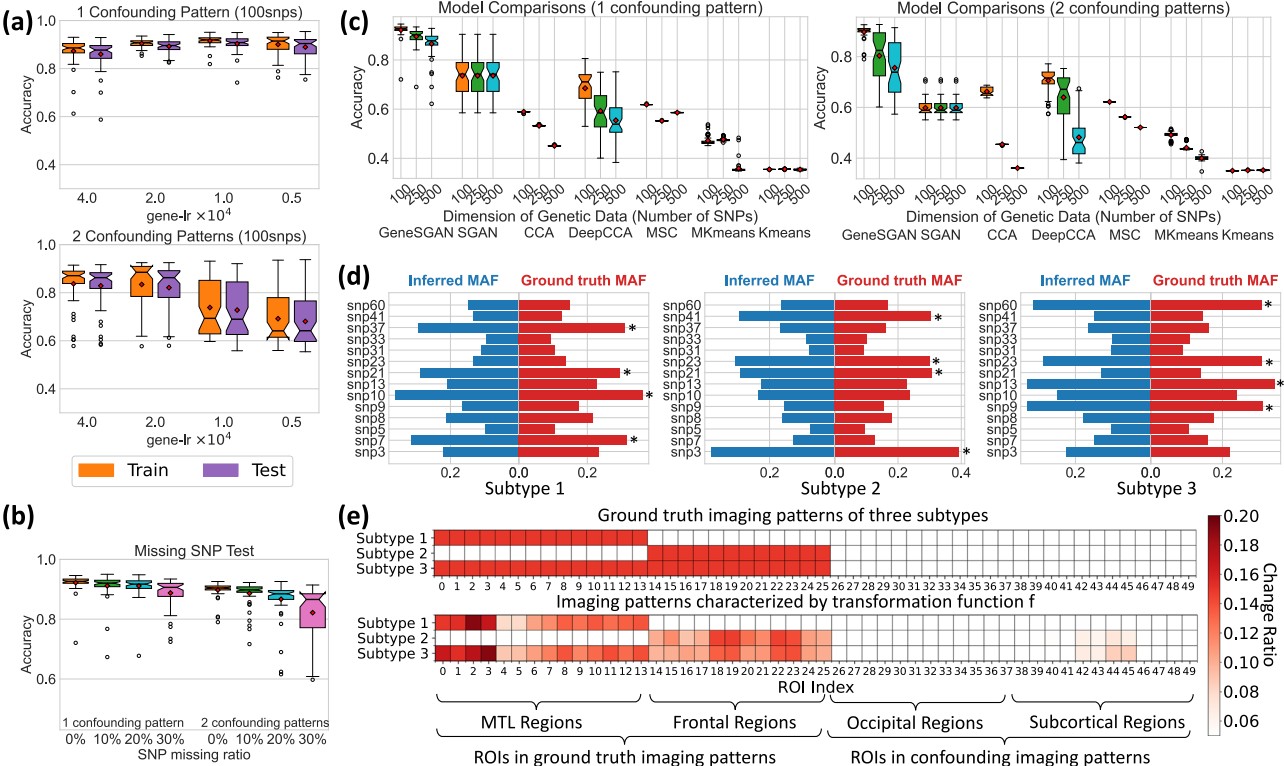

**Fig. 2 | Gene-SGAN identifies the ground truth in semi-synthetic experiments.** For constructing different ground truth subtypes, we impose distinct synthesized imaging patterns, specifically volumetric change in brain regions of interest (ROIs), on HC imaging features, simulating disease effects modulated by completely synthetic SNP variations. (Method 6). With known synthetic ground truth, we tested Gene-SGAN's clustering performance in several experimental scenarios: (**a**) generalizability, (**b**) robustness to missing genotype, (**c**) comparison to previous methods, and (**d**, **e**) interpretability for model performances. In (**a**), (**b**), (**c**), the box plots were generated from 50 datapoints that reveal clustering accuracies in 50-iteration of hold-out cross validation or model runs. **a** Gene-SGAN shows robust generalizability to test data. With different hyperparameter (gene-lr) settings, Gene-SGAN consistently achieves comparable clustering accuracies on the training and test sets. With increasing confounders in imaging features (bottom vs. top), achieving the model's optimal performance necessitates a higher gene-lr. **b** Gene-SGAN is robust to different levels of missing SNPs. Clustering accuracies remain high but gradually decrease as the SNPs' missing rate increases. **c** Gene-SGAN outperforms other clustering methods. We report the clustering performances of the seven models (Gene-SGAN vs. others) with different levels of simulated imaging confounders and dimensions of genetic data. SGAN: Smile-GAN; CCA: Canonical Correlation Analysis; MSC: Multiview-Spectral-Clustering; MKmeans: Multiview-KMeans. **d** Gene-SGAN accurately recovers SNPs' minor allele frequency (MAF) within each simulated subtype. We present the simulated subtype-associated SNPs (marked with asterisk) and the simulated confounding SNPs (not marked, Method 6). **e** Gene-SGAN captures dominant characteristics of the ground truth imaging patterns (associated with subtypes) but avoids confounding ones. The ROIs in the ground truth imaging patterns are colored with a ratio of 0.15, the average ratio of simulated changes (ranging from 0 to 0.3). The ROIs in the confounding patterns are left blank. Imaging patterns characterized by the model are defined as ratios of ROI changes made by the transformation function that approximates the disease effects. (Method 6) For visualizing important ROIs captured by f, we only color ratios > 0.05. MTL: medial temporal lobe. (Centerline, median; red marker: mean; box limits, upper and lower quartiles; whiskers, 1.5× interquartile range; points, outliers).

replaced 10%, 20%, and 30% SNPs with missing values, and retrained Gene-SGAN fifty times using all semi-synthetic participants with the optimal gene-lr selected through the CV procedure (Method 6). Gene-SGAN obtained good clustering performance in all cases, albeit with an expected, gradual drop in clustering performance with an increased missing rate (Fig. 2b).

### Gene-SGAN outperforms competing methods

With the known ground truth of the simulated subtypes, we compared the clustering performance of Gene-SGAN to six previously proposed methods (Method 6 and Supplementary Method 3): Smile-GAN model (SGAN, which shares the principal of weakly-supervised clustering with Gene-SGAN)[11], Canonical Correlation Analysis (CCA)[19], DeepCCA[20], Multiview-Spectral-Clustering (MSC)[21], Multiview-KMeans (MKmeans)[22], and Kmeans[23]. We ran each method fifty times using the complete datasets for different simulation cases, defined by varying levels of confounders and dimensions of genetic data. In all cases, Gene-SGAN outperformed these methods in terms of subtype assignment accuracies (Fig. 2c). The alternative methods were limited because they either could not incorporate genetic data or cluster

patient phenotypes directly, which can result in confounding by disease-unrelated variations, such as demographics and disease-unrelated genetic influences on the brain phenotypes. In contrast, Gene-SGAN is not only guided by both imaging and genetic data (multi-omics and multi-view) but also suppresses disease-unrelated confounding variations by effectively modeling mappings from HC to patient populations, which aims to cluster disease effects.

### Gene-SGAN accurately estimates simulated disease effects and genetic underpinnings, offering a means for mechanistic interpretations

Gene-SGAN offers a mechanism for interpreting identified subtypes and their related pathological processes. As an example, we utilized the fifty Gene-SGAN models trained for model comparisons on the dataset with 100 candidate SNPs and 2 confounding imaging patterns, leveraging outputs of functions h and f for interpreting the derived subtypes and model performances. (Method 6) The function h accurately inferred the simulated genetic distributions (MAFs of SNPs) of subtypes (Fig. 2d). Moreover, the derived transformation function f recovers the true simulated brain changes due to the subtype-specific

disease effects, elucidating the identified iAI signatures corresponding to each subtype. (Fig. 2e) Therefore, these functions of Gene-SGAN not only explain how the model determines the subtypes but also offer mechanistic insights into how the disease changes brain characteristics via genetically-mediated processes. (Method 1 and 6) Additionally, ancillary latent variables $z_2$ also accurately characterize two simulated confounding imaging patterns, proving the GeneSGAN's functionality in disentangling information and mitigating the impact of confounding factors (Supplementary Fig. 1).

## Subtypes of brain changes associated with AD and genetic variants

We first tested Gene-SGAN in the context of AD using 472 CN, 784 MCI, and 277 clinical AD participants from the ADNI study. We applied the Gene-SGAN model to the 144 imaging ROIs and 178 AD-associated SNPs of these participants, with $M$ being set as 3, 4, and 5. Gene-SGAN identified consistent and refined imaging subtypes from a coarse (e.g., $M = 3$) to a refined resolution (e.g., $M = 5$) (Supplementary Fig. 2 and Supplementary Note 1.3). The reproducibility of the identified subtypes was demonstrated through nested cross-validation, as well as experiments with external reference groups (Supplementay Method 7.1 and Supplementary Note 1.2). We presented the results with $M = 4$ because previous studies[11,14] consistently reported four distinct imaging subtypes in AD. Results for other resolutions of $M$ are presented in Supplementary Fig. 2a and Supplementary Data 1. We denoted the four subtypes related to AD as A1, A2, A3, and A4. Besides subtypes determined by the major latent variable $z_1$ of Gene-SGAN, we also examined $z_2$ and $z_3$ derived on this dataset, thereby validating their functionality in capturing non-linked imaging-specific and genetic-specific variations. (Supplementary Fig. 4).

## Subtypes related to AD show distinct imaging signatures

The participants assigned to each subtype showed distinct imaging patterns, as demonstrated in comparisons to cognitively normal HC participants (Fig. 3a) and in among-subtype comparisons (Fig. 3b). In voxel-based morphometry analyses, A1 ($N = 311$) exhibited relatively preserved regional brain volumes; A2 ($N = 197$) displayed focal medial temporal lobe (MTL) atrophy, prominent in the hippocampus and the anterior-medial temporal cortex; A3 ($N = 281$) showed widespread brain atrophy over the entire brain, including MTL; A4 ($N = 272$) displayed dominant cortical atrophy with relative sparing of the MTL.

## Subtypes related to AD show distinct genetic architectures

We tested SNP-subtype associations among 178 AD-associated SNPs using a likelihood-ratio test on two multinomial logistic regression models fitted with and without each SNP (Method 8), adjusting for covariates, including age, sex, *APOE ε4*, intracranial volume (ICV), and the first five genetic principal components. Through the test, we found that four subtypes are significantly different in 5 SNPs after Bonferroni correction ($p = 2.81 \times 10^{-4}$), including rs7920721 ($p = 1.1 \times 10^{-4}$), rs11154851 ($p = 3.2 \times 10^{-6}$), rs9271192 ($p = 4.2 \times 10^{-6}$), rs9469112 ($p = 2.7 \times 10^{-4}$), and rs4748424 ($p = 2.4 \times 10^{-4}$). Without controlling for *APOE ε4* as a covariate, rs429358 ($p = 7.8 \times 10^{-19}$) was the most significantly associated SNP – the strongest genetic risk factor in sporadic AD[24] (Fig. 3c and Supplementary Data 1). Detailed differences among subtypes in each SNP are demonstrated in Fig. 3c, which shows effective allele frequencies (EAF) within each subtype. A higher EAF indicates a higher risk of AD based on previous literature.

The effective incorporation of genetic features in Gene-SGAN significantly boosts the statistical power to detect robust SNP-subtype associations, as demonstrated by the comparison with the SmileGAN model[11], which derives four subtypes based on imaging features only using the similar weakly-supervised approach, yet the same test of SNP data within these subtypes does not identify any SNP-subtype associations on the same dataset after adjusting for *APOE ε4*.

## Subtypes related to AD show different clinical profiles

The four subtypes differed in age, sex, Aβ/tau measurements, white matter hyperintensity (WMH) volumes, and cognitive performance (Fig. 3d). Among participants diagnosed as MCI at baseline, A4 participants were the youngest group ($p < 0.001$ vs. all other groups) and included more females than A1 and A3 ($p = 0.004$ vs. A1 and $p < 0.001$ vs. A3). A3 participants had the most abnormal CSF Aβ ($p < 0.001$ vs. A1&A4 and $p = 0.039$ vs. A2). However, A2 participants showed significantly higher CSF p-tau than A3 participants ($p = 0.019$). For cognitive scores, A1 and A3 participants showed the best and the worst performances in memory, executive function, and language. A3-dominant participants also exhibited higher WMH volumes than all other dominant groups ($p < 0.001$ vs. A1&A4 and $p = 0.017$ vs. A2).

We also characterized the four subtypes with regard to additional 229 plasma and CSF biomarkers. Tested through one-way ANOVA, the four subtypes had significant differences in 18 plasma/CSF biomarkers after adjusting for multiple comparisons via Benjamini-Hochberg (B-H) method (Fig. 3e and Supplementary Data 2). In contrast, subtypes derived by Smile-GAN did not show significant differences in these biomarkers. The comparison suggests the increased power of Gene-SGAN in identifying subtypes that reflect heterogeneity not only in imaging and genetic features but also in other clinical biomarkers, implying greater 'biological relevance' for these subtypes. For example, among the associated biomarkers, tissue factor (TF) and von Willebrand factor (VWF) are highly expressed at blood-brain barrier[25,26], playing important roles in hemostasis; macrophage colony-stimulating factor (MCSF), CD40, chromogranin A (CgA), Cystatin-C contribute to microglial activation or proliferation[27-31]; angiotensin-converting enzyme (ACE) and heparin-binding EGF-like growth factor (HB-EGF) are involved in the process of Aβ degradation and clearance[32,33]. These results suggest potential biological pathways affected by AD disease mechanisms[34-36], which could be involved in disease pathogenesis or a direct or indirect response to disease-related neuroanatomical heterogeneity.

## Subtypes of brain changes associated with hypertension and genetic variants

In our second set of experiments with real data, we tested Gene-SGAN in the context of hypertension using 10,911 non-hypertensive and 16,414 hypertensive participants from the UKBB study. (Method 7) Hypertension has been a well-established risk factor for cerebrovascular diseases that contribute to dementia[18] via multiple potential processes, including atherosclerosis, small vessel ischemic disease, inflammation, and blood-brain barrier compromise. We used 144 imaging ROIs, total WMH volumes, and 117 hypertension-associated SNPs as features to train the model, with $M$ being set as 3, 4, and 5. Gene-SGAN identified consistent and refined imaging subtypes from a coarse (e.g., $M = 3$) to a refined resolution (e.g., $M = 5$) (Supplementary Fig. 3 and Supplementary Note 1.3). The reproducibility of the identified subtypes was demonstrated through both nested cross-validation and experiments involving independent reference or patient groups (Supplementary Method 7.2 and Supplementary Note 1.1). We denoted the five subtypes of brain changes associated with hypertension: H1, H2, H3, H4, and H5. Results for other resolutions of subtypes are presented in Supplementary Fig. 3a and Supplementary Data 3.

## Subtypes related to hypertension show distinct imaging signatures

Participants assigned to each subtype showed distinct imaging patterns compared to HC, non-hypertensive participants (Fig. 4a). H1 ($N = 4652$) participants showed mild atrophy in the midbrain. H2 participants ($N = 3543$) exhibited severe atrophy in subcortical gray matter (GM) regions as well as other white matter (WM) regions. H3 participants ($N = 5044$) showed larger volumes in deep structures of WM compared to HC. H4 participants ($N = 1341$) showed mild atrophy in

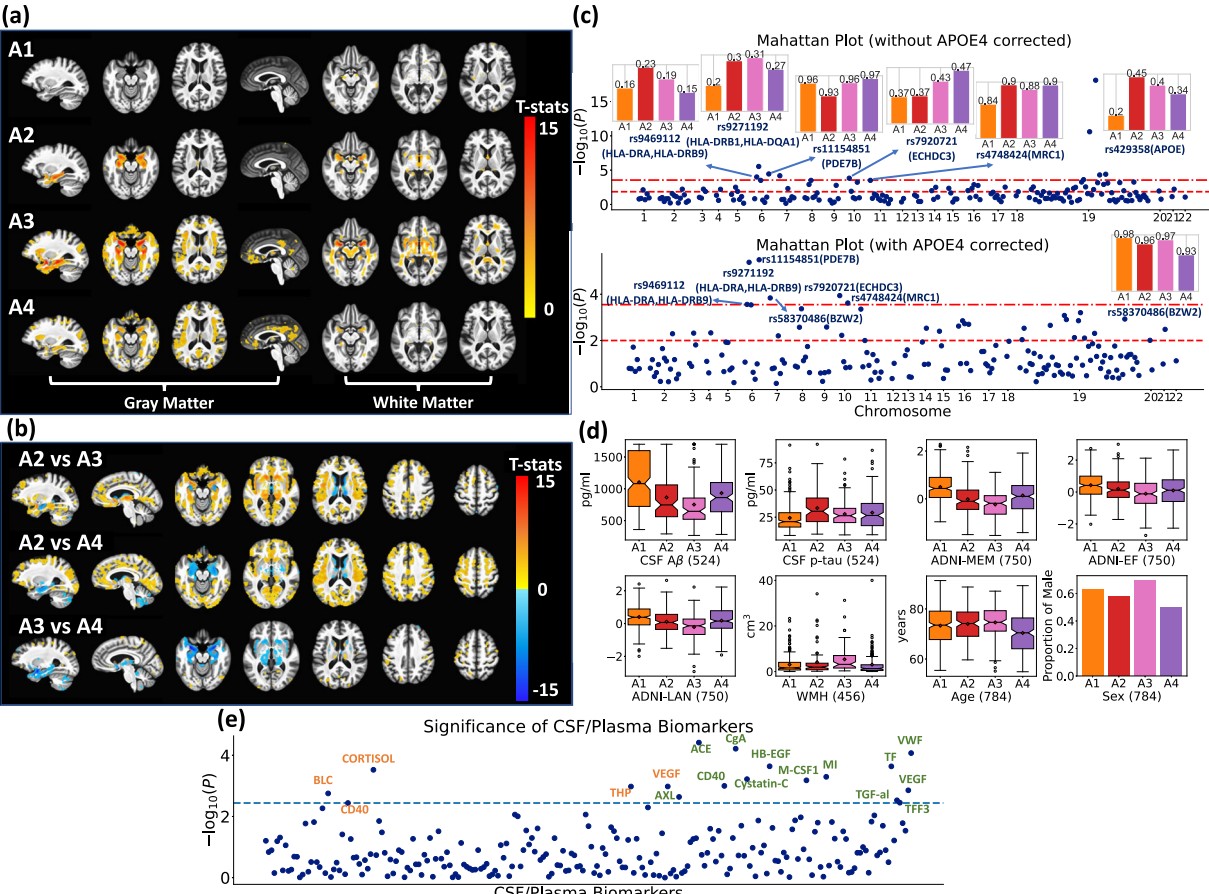

**Fig. 3 | Gene-SGAN identifies four subtypes of brain changes related to AD (A1, A2, A3, and A4). a** The four subtypes show different imaging patterns compared to HC. Warmer color denotes more brain atrophy in the subtype versus HC. **b** The four subtypes show distinct imaging patterns when compared with each other. In a comparison (subtype-i vs subtype-j), warmer color denotes relatively larger tissue volumes in subtype-i, and conversely for cooler color. In both (**a**, **b**), Voxel-wise group comparisons (two-sided t-test) were performed between two groups of participants. False discovery rate (FDR) correction was performed to adjust multiple comparisons with a *p*-value threshold of 0.05. **c** The four subtypes show distinct genetic underpinnings. The Manhattan plots show significant SNP-subtype associations among 178 AD-associated SNPs (one tailed likelihood-ratio test with multinomial logistic regression models) with (below) and without (above) adjusting for *APOE ε4*. The two dashed lines denote the *p*-value thresholds of 0.05 after adjusting for multiple comparisons via Bonferroni (top) and Benjamini-Hochberg (B-H, bottom) methods, respectively. We manually annotated the significant SNPs

that survived the Bonferroni correction with the SNP numbers and the mapped genes via their physical positions. We defined the effective allele of each SNP to be the allele positively associated with AD reported in previous literature. EAFs (effective allele frequencies) among each subtype are shown with bar plots. A higher frequency indicates a higher risk of AD. **d** The four subtypes show distinct clinical, cognitive, and demographic characteristics, including CSF Aβ and p-tau (other CSF biomarkers were separately evaluated in **e**). Box and whisker plots and bar plots reveal clinical, demographic, and cognitive variables of MCI participants by subtype. Sample sizes of each variable are presented beside their variable names. **e** The four subtypes show significant differences in CSF and plasma biomarkers. The Manhattan plots show the significance of differences (ANOVA test; one tailed test) among four subtypes related to AD. The dashed line represents the B-H corrected significance line. Orange-colored names: plasma biomarkers; green-colored names: CSF biomarkers. (Centerline, median; red marker: mean; box limits, upper and lower quartiles; whiskers, 1.5× interquartile range; points, outliers).

cortical and WM regions, as well as larger putamen, caudate, and higher WMH volumes. Finally, H5 (*N* = 1834) participants displayed widespread cortical atrophy in GM and WM with higher WMH volumes.

We further performed split-sampled analyses (Method 7) to test the replicability of imaging patterns associated with five subtypes. Patient data was evenly divided into the discovery set (*N* = 8207) and the replication set (*N* = 8207). The Gene-SGAN model, retrained on the discovery set, was applied to both sets to re-derive the five hypertension-related subtypes. Highly consistent imaging signatures were observed between discovery and replication sets (Supplementary Fig. 5), which also have strong agreements with the subtypes derived from the entire dataset (Fig. 4a).

## Subtypes related to hypertension show distinct genetic architectures

We first tested SNP-subtype associations among 117 hypertension-associated SNPs and 5 subtypes using all available hypertensive patient

data (*N* = 16,414). 27 SNPs were identified to be significantly different among subtypes after Bonferroni correction for multiple comparisons (Fig. 4c and Supplementary Data 3). We further examined the replicability of SNP-subtype associations in split-sampled analyses (Method 7 and Supplementary Method 4). Among the discovery set, we found 15 significant SNP-subtype associations (Method 8) after Bonferroni correction. Among these, 10 SNP-subtype associations (66.7%) were replicated in the replication set at B-H corrected significance, and 7 (46.7%) were significant after Bonferroni corrections for multiple comparisons (Supplementary Method 4).

In contrast, we found only 5 SNPs significantly associated with Smile-GAN subtypes in the discovery set. Among them, only 1 SNP was replicated in the replication set at both B-H corrected and Bonferroni corrected significance (Supplementary Data 5). Details of the reproduced SNPs are presented in Supplementary Data 4. These results further support the increased power of Gene-SGAN in GWAS.

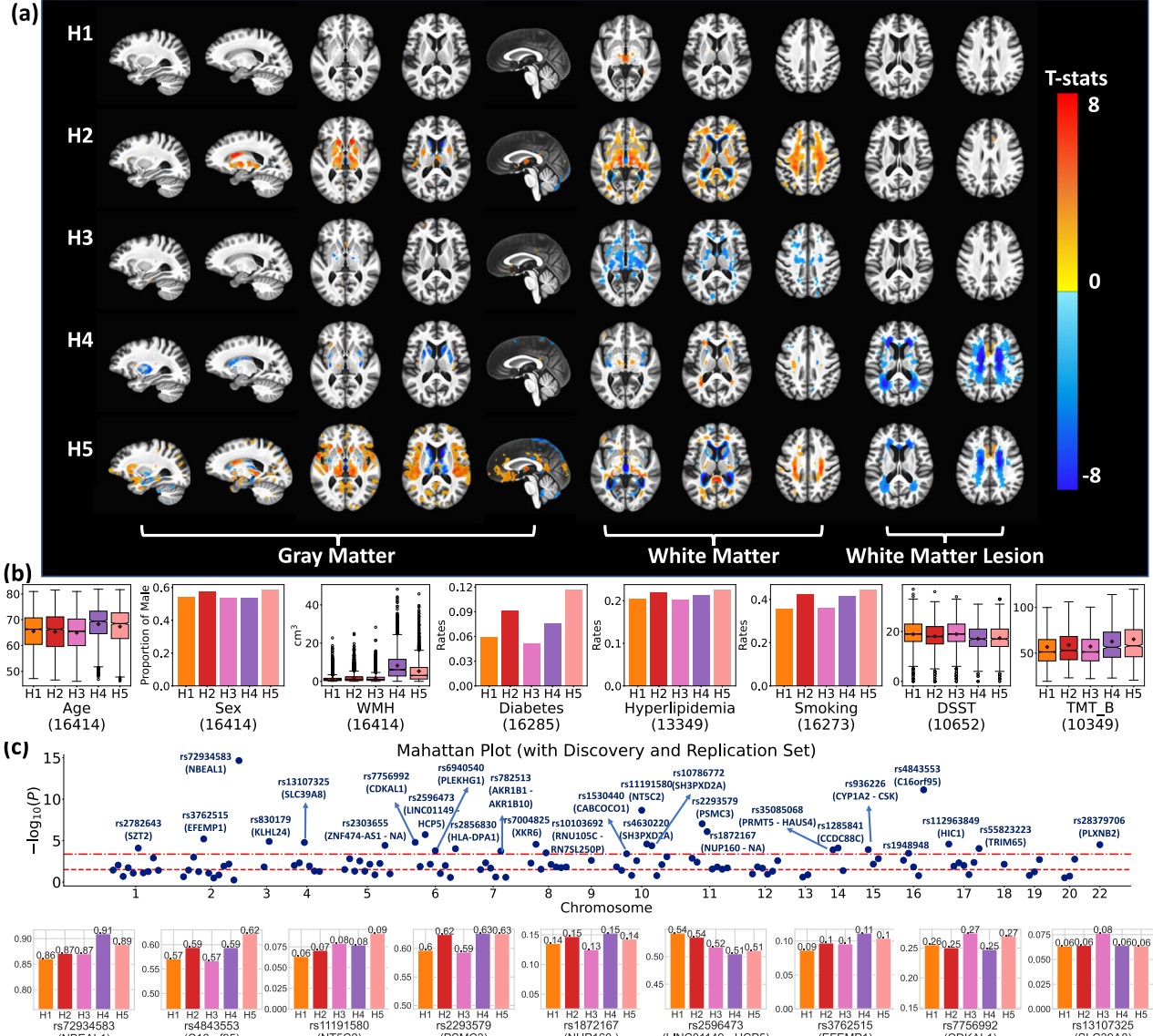

**Fig. 4 | Gene-SGAN identifies five subtypes of brain changes related to hypertension (H1-H5). a** The five subtypes show distinct imaging patterns. Voxel-wise group comparisons (two-sided t-test) were performed between HC participants (i.e., non-hypertensive participants) and participants assigned to H1, H2, H3, H4, and H5, respectively. False discovery rate (FDR) correction for multiple comparisons with a *p*-value threshold of 0.05 was applied. Warmer color denotes brain atrophy (i.e., HC > subtype), and cooler color represents larger tissue volume (i.e., subtype > HC). **b** The five subtypes show distinct clinical, cognitive, and demographic characteristics. Box and whisker plots and bar plots show the characteristic of demographic, clinical, and cognitive variables. In TMT B plots, outliers are excluded for visualization purposes. (center line, median; red marker: mean; box limits, upper and lower quartiles; whiskers, 1.5× interquartile range; points,

outliers). Sample sizes of each variable are presented below each figure. **c** The five subtypes show different genetic architectures. The Manhattan plot displays significant SNP-subtype associations among 117 hypertension-associated SNPs (one tailed likelihood-ratio test with multinomial logistic regression models) using all hypertensive patients' data. The two dashed lines denote the p-value thresholds of 0.05 after adjusting for multiple comparisons via Bonferroni (top) and B-H methods (bottom). We manually annotated significant SNPs that survived the Bonferroni correction with the SNP numbers and the mapped genes via their physical positions. In this figure, we defined the effective allele of each SNP as the allele positively associated with hypertension or WMH (SNPs identified in hypertension-gene interaction analyses[54]) reported in previous literature. The bar plots reveal EAFs of the top nine significant SNPs within each subtype.

## Subtypes related to hypertension show distinct clinical profiles

We examined the clinical profiles of the five subtypes with regard to demographics, comorbidities, and cognitive scores (Fig. 4b). WMH volumes were the highest among H4 participants (*p* < 0.001 vs. all other H) and the second highest among H5 participants (*p* < 0.001 vs. H1-H3). In terms of comorbidities, H5 participants had significantly higher rates of diabetes (*p* < 0.001 vs. H1, H3, H4, and *p* = 0.004 vs. H2), while H2 and H4 participants also displayed significantly higher rates of diabetes than H1 and H3 participants (*p* < 0.001 for H2 vs. H1&H3, *p* = 0.039 for H4 vs. H1, and *p* = 0.001 for H4 vs. H3). The five subtypes did not exhibit significant differences in hyperlipidemia. H1 and H3

participants had a significantly smaller proportion of smokers (*P* < 0.001 vs. all other groups). In cognition, H4 and H5 participants showed the worst performance as measured by DSST and TMT-B (*p* < 0.001 vs. H1-H3 in three scores), while H2 participants showed worse performance than H1 and H3 participants (*p* < 0.001 in DSST and TMT-B).

## Discussion

In the current work, we proposed Gene-SGAN – a novel deep weakly-supervised clustering method – to unravel disease heterogeneity and develop genetically-explained disease subtypes having distinct brain

phenotypes. The novelty of Gene-SGAN lies in its multi-view modeling nature, which ensures that the derived disease subtypes not only reflect distinct neuroanatomical patterns but are also associated with underlying genetic determinants. The following two central points bolster the advantage of Gene-SGAN over other clustering methods[11,19–23]. First, through the generative and weakly-supervised modeling of disease heterogeneity, Gene-SGAN seeks to cluster based on disease-related variations in phenotypic data, achieved via clustering of transformations of brain phenotypes, which helps avoid confounding variations associated with demographic and disease-irrelevant genetic factors. Second, Gene-SGAN links phenotypic and genetic variations through three sets of latent variables, which separately encode linked and non-linked genetic and phenotypic variations, thereby enabling the identification of unbiased phenotypic subtypes with genetic associations. Through extensive and systematic semi-synthetic and real data experiments, we demonstrated the efficacy and applicability of Gene-SGAN in deriving biologically and clinically distinct disease subtypes.

Our experiments focused on applying Gene-SGAN to imaging features from brain MRIs and SNPs. However, Gene-SGAN is widely applicable to other multi-omics biomedical data, including various sources of phenotypic (e.g., other clinical variables) and genetic features (e.g., gene expression data). Specifically, with preselected reference populations and genetic features, Gene-SGAN can effectively discover genetically-associated subtypes of phenotypic features related to various diseases or disorders. Moreover, commonly observed missing genotypes limit the applicability of many multivariate methods that require dropping participants with missing features. In contrast, Gene-SGAN is robust to missing genotypes and optimally models disease heterogeneity with all available genetic features. This property is essential in imaging genomics studies, which often suffer from relatively small sample sizes due to the difficulty in data collection. The high tolerance and sophisticated adoption of missing data enable Gene-SGAN to be a general method for modeling disease heterogeneity in biomedical studies. In our experiments, we demonstrated the robust applicability of Gene-SGAN by applying it to two independent datasets for studying two distinct pathologies: AD and hypertension. The method effectively discovers genetically and neuroanatomically associated subtypes in AD/MCI and hypertension, yielding increased statistical power for downstream statistical analyses.

Gene-SGAN identified four AD-related subtypes with distinct characteristics. A1 is characterized by preserved brain volume with the lowest levels of cognitive impairment and Aβ/tau deposition, indicating a resilient subtype. A2 is associated with focal MTL atrophy and high CSF-tau, suggesting a subtype driven by limbic-predominant, likely rapidly progressive neuropathology (high tau levels, despite the localized nature of neurodegeneration). A3 is characterized by severe atrophy in cortical and MTL regions, the most abnormal CSF-Aβ, the worst cognitive performances, and the highest WMH volumes. Patients assigned to this subtype might mainly have a manifestation of typical AD pathology as well as vascular co-pathology. A4 participants exhibit severe cortical but relatively modest MTL atrophy patterns, indicating a mixture of participants with a cortical presentation of AD and those with other neural degenerative processes (e.g., advanced brain aging[4]). A significantly lower age range of A4 participants suggests the inclusion of EOAD participants, who were characterized by hippocampal-sparing disease with posterior cortical atrophy[37]. Notably, the four subtypes were significantly associated with known AD-related genetic variants. Among all subtype-associated SNPs, rs429358 in the *APOE* gene was the strongest genetic risk factor for sporadic AD[24] and was associated with hippocampal atrophy and cognitive decline[38,39]. Two other SNPs (rs9469112 and rs9271192) were mapped to the HLA region that was involved in immune response modulation[40–42]. Our results showed a lower frequency of EAFs of these three SNPs among A1

participants, indicating a protective effect contributing to the observed resilience. In contrast, the highest EAFs of rs429358 and significant MTL atrophy of A2 participants resemble the previously reported characteristics of limbic-predominant AD[43]. Higher EAFs in rs9469112 and rs9271192 in A2 and A3 suggest the potential inflammatory mechanisms contributing to these two subtypes. A4 participants have the highest EAF in rs7920721, an SNP exclusively associated with AD among participants who don't carry *APOE ε4*[44]. Our A4 subtype supports this finding while further linking the effect of this SNP with an atypical atrophy pattern of AD, likely accompanied by co-pathologies.

The identified AD-related subtypes exhibit similarities to other neuroimaging-based clustering studies, including the identification of MTL- and cortical-predominant patterns[11,13,14,45]. However, Gene-SGAN's subtypes were specifically refined to maximize genetic associations. As such, it focuses on deriving imaging endophenotypes, rather than just phenotypic clusters. For instance, we observed two extreme subtypes characterized by highly focal hippocampal atrophy and preserved brain volume, showing significantly differences in rs429358 (*APOE*) with the highest and lowest EAFs, respectively. The minimal differences in *APOE ε2* and *APOE ε4* among the other neuroimaging-based subtypes[13,14,45] indirectly verify the refinements provided by Gene-SGAN. Moreover, in direct comparisons to the subtypes presented in ref. 11, Gene-SGAN's subtypes demonstrate much stronger genetic associations, further validating the effectiveness of refinements. It is worth noting that certain previously identified atrophy subtypes, such as occipital atrophy patterns[45], are primarily captured by latent variables $z_2$, suggesting their limited genetic associations. Critically, compared to AD-related subtypes derived with imaging features only[11], these four subtypes possess more significant differences in a large set of plasma/CSF biomarkers, which are related to distinct biological mechanisms contributing to the heterogeneity of AD. Taken together, the four AD-related subtypes identified by Gene-SGAN support the conclusion that disease heterogeneity modeling with genetic guidance better reflects upstream biological processes and boosts downstream analyses' statistical power.

Gene-SGAN identified five clinically distinct hypertension-related subtypes, which reflect the remarkable heterogeneity of the effects of hypertension on brain structure. The H1 and H3 participants exhibit the best cognitive performances and lowest rates of comorbidities. Though sharing similarities in preserved GM volumes, these subtypes differ in WM structures. The H2 subtype is characterized by subcortical and WM atrophy, with a higher rate of diabetic participants. Previous studies have reported hypertension-related influence on subcortical morphology[46] and WM integrity[47,48]. However, WM microstructures of these subtypes need to be further explored through diffusion MRI. Both the H4 and H5 subtypes are associated with high WMH volumes, the most commonly used biomarker of cerebral small vessel ischemic disease, and worse cognitive performances. In addition, a higher rate of diabetes is observed among H5 participants, which partially explains the widespread atrophy patterns associated with the H5 subtype based on previous studies[49,50]. These five subtypes not only resemble the previously reported associations among blood pressure, comorbidities, and neuroanatomical changes[47–53], but also further dissect variations in brain changes, potentially elucidating heterogeneous effects from various underlying hypertension-related or hypertension-inducing pathological processes. Among subtype-associated SNPs, rs72934583, rs4843553, and rs3762515 were previously associated with WMH volumes[54], which are consistent with their higher EAFs among H4 and H5 participants in our findings. rs11191580, rs7756992, and rs13107325 were linked in previous GWAS to diabetes and obesity[55–58], two comorbidities of hypertension.

We demonstrated the generalizability of Gene-SGAN clustering through cross-validation and independent replication. The

reproducibility crisis[59] has drawn much attention in machine learning and casts a shadow over future clinical translation. For genetic studies, replication of associations also underpins the reliability of discovered genetic variants with modest effect sizes[60]. First, extensive cross-validated experiments on semi-synthetic datasets supported the generalizability of Gene-SGAN's performance to unseen data in identifying simulated subtypes. Second, reproducible subtypes and latent variables could be identified on real datasets using cross-validation, different model scales (M), as well as completely independent reference/target groups. Furthermore, we validated Gene-SGAN's ability to identify subtypes with replicable genetic associations on real datasets through split-sampled experiments on the hypertensive population. Several methodological considerations ensure the reproducibility of Gene-SGAN subtypes. For example, the incorporation of ancillary latent variables (e.g., $z_2$ and $z_3$) and sparse transformations implicitly regularized SNP-subtype associations identified in the discovery set. The ensemble learning procedure for deriving the final subtypes (e.g., the consensus of models derived through hold-out CV) further encouraged the replicability of SNP-subtype associations. In conclusion, Gene-SGAN can derive reproducible subtypes that are also biologically and genetically interpretable.

The potential clinical impact of Gene-SGAN is versatile. In general, it helps dissect the heterogeneity of diseases into relatively more neuroanatomical subtypes that also have genetic underpinnings, and hence it contributes to precision diagnostics that can have downstream effects on any subsequent analysis. For example, deriving robust disease-related subtypes may help improve classification performance for individualized disease diagnosis and prognosis. Moreover, modeling disease heterogeneity provides new patient stratification and treatment evaluation tools for future clinical trials, which remain important in the setting of mixed results and clinical limitations of anti-amyloid treatments. It is well recognized that evaluating treatment responses within relatively more homogeneous subgroups of patients can significantly increase the power of clinical trials. Our results also suggest that disease subtyping via Gene-SGAN could augment our ability to detect significant imaging and genomic characteristics of AD, which would be diluted in case-control comparisons due to the underlying heterogeneity. Finally, Gene-SGAN subtypes are genetically relevant by modeling, which serves as a reliable endophenotype to pinpoint potential causal genetic variants for drug repurposing and discovery.

There are potential improvements to the current study. First, the MCI/AD subtypes were derived from a relatively modest sample size ($N = 1061$). Ongoing efforts, including the unprecedented consolidation of large-scale imaging-genomic consortia of AD, such as the AI4AD consortium (http://ai4ad.org/), may provide opportunities to produce more reproducible and diverse disease subtypes, including less common genetic-structural patterns. Second, though Gene-SGAN identified consistent subtypes across different model scales, promoting a hierarchical relationship among multiscale clustering results could potentially provide better interpretations of the identified subtypes in certain scenarios. Third, in this study, we validated the performance of Gene-SGAN using candidate SNPs directly associated with the disease of interest. For future applications of Gene-SGAN, it might deserve to try different SNP selection criteria that restrict or relax the scope of candidate SNPs. For instance, we could incorporate SNPs based on information from druggable genes[61], thereby providing more insights into drug discoveries.

In summary, Gene-SGAN effectively unravels phenotypic variations associated with genetic factors into multiple disease-related subtypes to comprehensively understand disease heterogeneity. Gene-SGAN can be widely and easily applied to biomedical data from different sources to derive clinically meaningful disease subtypes. These iAI subtypes provide great potential for drug discovery and repurposing, optimization of clinical trial recruitment, and personalized medicine based on an individual's genetic profile.

## Methods

### Method 1. The Gene-SGAN model

Gene-SGAN is a multi-view, deep weakly-supervised clustering method based on GAN and variational inference (VI). Gene-SGAN aims to cluster patients from varying sources of phenotypic (e.g., clinical variables or imaging features derived from MRI) and genetic features (e.g., SNP). For this purpose, the model learns three sets of latent variables. The M-dimensional categorical variable, comprising the vector $z_1$, captures joint genetic and phenotypic variations and indicates the probabilities of M output clusters, referred to as subtypes. Two ancillary sets of variables, comprising the vectors $z_2$ and $z_3$, summarize phenotypic-specific and genetic-specific variations, respectively. Critically, the model avoids confounders from disease-unrelated variations in phenotypic features under the framework of weakly-supervised clustering[62]. To sum up, Gene-SGAN employs a GAN generative model to construct one-to-many mappings from a reference population (i.e., HC) to a target population (i.e., patient) instead of clustering based on global similarity/dissimilarity in the patient population, which might be affected by demographics, lifestyle, or disease-unrelated genetic influences. In addition, VI is used to ensure that disease-related genetic features jointly guide the clustering solution. The framework of Gene-SGAN (Fig. 1b) consists of two main optimization steps: the Phenotype step (via GAN) and the Gene step (via VI), as detailed below.

For conciseness, we denote the following variables: $x$: the REF phenotypic features; $y$: the TAR phenotypic features; $y'$: the synthesized TAR phenotypic features; $v$: the genetic features; $z_1, z_2, z_3$: the three latent variables in Gene-SGAN. The distributions of these variables are denoted as: $x \sim p_{ref}(x)$, $y \sim p_{tar}(y)$, $y' \sim p_{syn}(y')$, $z_1 \sim p_{\theta_{z_1}}(z_1)$, $z_2 \sim p_{z_2}(z_2)$, $z_3 \sim p_{z_3}(z_3)$, where $p_{\theta_{z_1}}(z_1)$ is a parametrized distribution updated during the training procedure. In addition, $\theta_D, \theta_f, \theta_g, \theta_h, \theta_r$, represent parameters of the five parametrized functions (Fig. 1b).

**Phenotype step.** The Phenotype step incorporates only the phenotypic features for optimization. Gene-SGAN learns one transformation function, $f : X \times Z_1 \times Z_2 \to Y$, that maps REF features $x$ to different synthesized TAR features $y' = f(x, z_1, z_2)$, with the latent variables $z_1, z_2$ specifying variations in the synthesized features (i.e., different mapping directions). Specifically, $z_1$ contributes to phenotypic variations with genetic associations. In contrast, $z_2$ contributes to phenotype-specific variations without genetic associations. The latent variable, $z_2 \sim p_{z_2}(z_2)$, is sampled from a predefined multivariate uniform distribution $U[0,1]^{n_{z_2}}$ with the dimension $n_{z_2}$; $z_1 \sim p_{\theta_{z_1}}(z_1)$ follows an M-dimensional categorical distribution (i.e., an M-dimensional one-hot vector) parametrized by $\theta_{z1}$ (i.e., probabilities of categories). This enables the model to derive robust clustering solutions with imbalanced cluster sizes. An adversarial discriminator D is introduced to distinguish between the real TAR features $y$ and the synthesized TAR features $y'$, thereby ensuring that the synthesized TAR features from f are indistinguishable from the real TAR features. Moreover, we introduce three types of regularization to encourage that the transformation function approximates underlying pathological processes: the sparse transformation, the Lipschitz continuity, and the inverse consistency. The complete objective function of the Phenotype step is thus a combination of the adversarial loss and the regularization terms, as detailed below.

First, we modify the adversarial loss of the basic GAN and the Smile-GAN model to update the distribution of the latent variable $z_1 \sim p_{\theta_{z_1}}(z_1)$, so that the clustering model is robust to imbalanced data. Details of this modification are presented in Supplementary

Method 1.2. The modified adversarial loss is written as:

$$L_{GAN}(\theta_D,\theta_f,\theta_{z_1}) = E_{y \sim p_{tar}(y)}\big[\log(D(y))\big] + E_{y' \sim p_{syn}(y')}\big[1 - \log(D(y'))\big]$$
$$+ \kappa D_{KL}(p_U(z_1)|p_{\theta_{z_1}}(z_1)) \tag{1}$$

$$= E_{z_1 \sim p_{\theta_{z_1}}(z_1), y \sim p_{tar}(y)}\big[\log(D(y))\big]$$
$$+ E_{z_1 \sim p_{\theta_{z_1}}(z_1), z_2 \sim p_{z_2}(z_2), x \sim p_{ref}(x)}\big[1 - \log(D(f(x,z_1,z_2)))\big] \tag{2}$$
$$+ \kappa D_{KL}\big(p_U(z_1)|p_{\theta_{z_1}}(z_1)\big)$$

$$= M * E_{y \sim p_{tar}(y), z_1 \sim p_U(z_1)}\big[p_{\theta_{z_1}}(z_1)\log(D(y))\big]$$
$$+ M * E_{z_1 \sim p_U(z_1), z_2 \sim p_{z_2}(z_2), x \sim p_{ref}(x)}\big[p_{\theta_{z_1}}(z_1)(1 - \log(D(f(x,z_1,z_2))))\big]$$
$$+ \kappa D_{KL}\big(p_U(z_1)|p_{\theta_{z_1}}(z_1)\big)$$

$$\tag{3}$$

Intuitively, we sample $z_1$ from a discrete uniform distribution, $p_U(z_1)$, but penalize the losses with its probability under the distribution $p_{\theta_{z_1}}(z_1)$. The modified loss function enables $z_1$ to be implicitly sampled from the parameterized distribution $p_{\theta_{z_1}}(z_1)$. $p_{\theta_{z_1}}(z_1)$ is controlled to be not far away from $p_U(z_1)$. Both $\theta_f$ and $\theta_{z_1}$ are optimized so that the synthesized TAR features follow similar distributions as the real TAR features. The discriminator D provides a probability – **y** comes from the real features rather than the generator – and aims to distinguish the synthesized TAR features from the real TAR features. Therefore, the discriminator attempts to maximize the adversarial loss function while $\theta_f$ and $\theta_{z_1}$ are updated to minimize it. The corresponding training process can be denoted as:

$$\min_{\theta_f,\theta_{z_1}} \max_{\theta_D} L_{GAN}\big(\theta_D,\theta_f,\theta_{z_1}\big) = E_{y \sim p_{tar}(y)}\big[\log(D(y))\big]$$
$$+ E_{y' \sim p_{syn}(y')}\big[1 - \log(D(y'))\big] + \kappa D_{KL}\big(p_U(z_1)|p_{\theta_{z_1}}(z_1)\big) \tag{4}$$

Second, we assume that disease-related processes primarily affect a subset of phenotypic features (i.e., sparsity). We, therefore, define a change loss to be the $l_1$ distance between the synthesized TAR features and the REF features to boost sparse transformations:

$$L_{change}(\theta_f) = E_{x \sim p_{ref}(x), z_1 \sim p_U(z_1), p_{z_2}(z_2)}\big[||f(x,z_1,z_2) - x||_1\big] \tag{5}$$

Third, the inverse consistency is accomplished by introducing an inverse mapping function g for re-estimating $z_1$ and $z_2$ from the synthesized TAR features $f(x,z_1,z_2)$. For clarity in description, we define $g_1$ and $g_2$ as two inverse mapping functions that re-estimate $z_1$ and $z_2$, respectively, though they share the same encoding neural network. The cross-entropy loss is used for reconstructing the categorical variable $z_1$, While the $l_2$ loss is used for the continuous variable $z_2$. By denoting $l_c$ to be the cross-entropy loss with $l_c(a,b) = -\sum_{i=1}^k a^i \log b^i$, we define the reconstruction loss as:

$$L_{recons}(\theta_f,\theta_{g_1},\theta_{g_2}) = E_{x \sim p_{ref}(x), z_1 \sim p_U(z_1), z_2 \sim p_{z_2}(z_2)}\big[l_c(z_1,g_1(f(x,z_1,z_2)))\big]$$
$$+ E_{x \sim p_{ref}(x), z_1 \sim p_U(z_1), z_2 \sim p_{z_2}(z_2)}\big[||g_2(f(x,z_1,z_2)) - z_2||_2\big] \tag{6}$$

The minimization of the reconstruction loss enables the transformation function, f, to identify sufficiently distinct features in the TAR domain depending on the latent variables, $z_1$ and $z_2$[11]. Moreover, the inverse function $g_1$ is critical in the model framework. First, it is optimized in the Gene step (detailed below) for inferring the distributions of the genetic features, thereby allowing the clustering solutions to be genetically guided. Moreover, it serves as a clustering function after training to estimate $z_1$ from real TAR phenotypic

features, deriving the probabilities of the cluster memberships (i.e., subtypes). More details of the inverse functions are stated at the end of this section and in Supplementary Method 1.1.

Lipschitz continuities of the functions f, $g_1$, $g_2$ are ensured through weight clipping as described in Supplementary Method 2.2 instead of through additional loss functions. Therefore, the full objective of the Phenotype step can be written as:

$$L_{Phenotype}\big(\theta_D,\theta_f,\theta_{g_1},\theta_{g_2},\theta_{z_1}\big) = L_{GAN}\big(\theta_D,\theta_f,\theta_{z_1}\big) + \mu L_{change}(\theta_f)$$
$$+ \lambda L_{recons}\big(\theta_f,\theta_{g_1},\theta_{g_2}\big) \tag{7}$$

where μ and λ are two hyperparameters that control the relative importance of each loss function during the training process. Through this objective, we aim to derive parameters, $\theta_D,\theta_f,\theta_{g_1},\theta_{g_2},\theta_{z_1}$, such that:

$$\theta_D,\theta_f,\theta_{g_1},\theta_{g_2},\theta_{z_1} = \arg \min_{\theta_f,\theta_{g_1},\theta_{g_2},\theta_{z_1}} \max_{\theta_D} L_{Phenotype}\big(\theta_D,\theta_f,\theta_{g_1},\theta_{g_2},\theta_{z_1}\big) \tag{8}$$

**Gene step.** Different from the Phenotype step, we do not include the reference group for learning a transformation of genetic features due to their innateness and immutability. Also, we opt not to incorporate a feature selection mechanism with respect to reference data within the model framework, as it requires a large dataset to comprehensively identify disease-associated SNPs. Instead, we pre-select candidate SNPs associated with the disease of interest using the GWAS-Catalog[63] online portal. The Gene step encourages the clustering solution of the target group to be associated with the candidate genetic features. This approach not only provides more comprehensive selection of candidate SNPs but also makes Gene-SGAN more available to users lacking large imaging-genomic datasets.

Specifically, the model learns a parametrized distribution of the genetic features **v**, conditioned on the phenotypic features **y** and a new latent variable $z_3$, $p_{\theta_h,\theta_{g_1}}(v|z_3,y)$, where $z_3$ characterizes genetic-specific variations unrelated to the phenotypic features. This is accomplished through the VI method for approximating an intractable posterior distribution, $p_{\theta_h,\theta_{g_1}}(z_3|v,y)$ by a variational distribution $q_{\theta_r}(z_3|v,y)$. From the KL divergence between $p_{\theta_h,\theta_{g_1}}(z_3|v,y)$ and $q_{\theta_r}(z_3|v,y)$, we can derive the evidence lower bound (ELBO) for $p(v|y)$. The derivation is presented in Supplementary Method 1.3.

$$\log p(v|y) \geq E_{z_3 \sim q_{\theta_r}(z_3|v,y)}\left[\log p_{\theta_h,\theta_{g_1}}(v|z_3,y) + \log \frac{p_{z_3}(z_3)}{q_{\theta_r}(z_3|v,y)}\right] \tag{9}$$

The conditional distribution $p_{\theta_h,\theta_{g_1}}(v|z_3,y)$, is parametrized by the functions h and $g_1$. Based on different sources of genetic features, it can be modeled as different types of distributions. For instance, herein, we use SNPs as genetic features. We define $p_{\theta_h,\theta_{g_1}}(v|z_3,y) = B(2,p_{binom}^{n_{genetic}})$ as a multivariate binomial distribution, which has the number of trials equaling two and the dimension equaling the number of SNPs ($n_{genetic}$). The function h takes $z_3$ and $g_1(y)$ as inputs and outputs parameters of the conditional distribution, $p_{\theta_h,\theta_{g_1}}(v|z_3,y)$ (e.g., the parameters, $p_{binom}^{n_{genetic}}$, represent MAFs of SNPs when using SNPs as features). The variational distribution, $q_{\theta_r}(z_3|v,y) = N(\mu^{n_{z_3}},\sigma^{n_{z_3}})$, parametrized by a function r, is modeled as a multivariate normal distribution with the dimension equaling $n_{z_3}$. The function r takes **v** and **y** as inputs and outputs $\mu$ and $\sigma$ for each dimension of $z_3$. The prior distribution of $z_3$, $p_{z_3}(z_3) = N(0^{n_{z_3}},1^{n_{z_3}})$, is defined as a multivariate standard distribution. In the Gene step, we

maximize the ELBO for p(**v**|**y**) by minimizing the following function:

$$-E_{z_3 \sim q_{\theta_r}(z_3|v,y)}\left[\log p_{\theta_h,\theta_{g_1}}(v|z_3,y) + \log \frac{p_{z_3}(z_3)}{q_{\theta_r}(z_3|v,y)}\right] \quad (10)$$

In the case of missing genetic features, we substitute the missing features with the mean value over observed features among the target population (e.g., two times MAF within the target population for SNP data) and use the imputed genetic features, $v^{impute}$, as the inputs for the function r. The conditional distribution, $p_{\theta_h,\theta_{g_1}}(v|z_3,y)$, in ELBO is computed with only the observed genetic features $v^{observe}$[64]. Therefore, the objective function of the Gene step is written as:

$$L_{Gene}\left(\boldsymbol{\theta_h},\boldsymbol{\theta_{g_1}},\boldsymbol{\theta_r}\right) = -E_{z_3 \sim q_{\theta_r}(z_3|v^{impute},y)}$$
$$\left[\log p_{\theta_h,\theta_{g_1}}\left(v^{observe}|z_3,y\right) + \log \frac{p_{z_3}(z_3)}{q_{\theta_r}(z_3|v^{impute},y)}\right] \quad (11)$$

Through this objective function, we derive $\boldsymbol{\theta_h},\boldsymbol{\theta_{g_1}},\boldsymbol{\theta_r}$ such that

$$\boldsymbol{\theta_h},\boldsymbol{\theta_{g_1}},\boldsymbol{\theta_r} = \arg\min_{\theta_h,\theta_{g_1},\theta_r} L_{Gene}\left(\boldsymbol{\theta_h},\boldsymbol{\theta_{g_1}},\boldsymbol{\theta_r}\right) \quad (12)$$

Notably, the term, $\log\frac{p_{z_3}(z_3)}{q_{\theta_r}(z_3|v^{impute},y)}$, can also be considered a regularization term. Through regularization, we control the contribution of $z_3$ in the inference of genetic distributions and thus guarantee the contribution from the phenotypic features **y**.

The Phenotype and Gene optimization steps are performed iteratively during the training process. The learning rate of the Gene step (i.e., gene-lr) controls the weight on genetic features during the training procedure, serving as a hyperparameter for different cases (Result). Other implementation details of the model, including network architectures, training details, algorithm, and training stopping criteria, are presented in Supplementary Method 2.

**Subtype assignment.** After the training process, the clustering function $g_1$ can be applied to the training and independent test patient data to estimate the latent variable $z_1$ that indicates the subtypes of interest (i.e., categorical subgroup membership). Specifically, $g_1$ outputs $M$ probability values ($P_i$) for each participant, with each probability corresponding to one subtype and the sum of $M$ probabilities being 1 ($\sum_{i=1}^{M} P_i = 1$). We then assign each participant to the dominant subtype, determined by the maximum probability (subtype = $\arg\max_i P_i$). Notably, subject-level reference data is not directly necessary for trained model applications. Only the parameters estimated from the reference group are used for PT input feature standardization. We could directly use the stored parameters estimated from the training reference group, while the impact of standardization with respect to a

new reference group was analyzed in Supplementary Figure 1 and Supplementary Note 1.

## Method 2. Study populations

MRI (Method 3) and clinical (Method 4) data used in this study were consolidated and harmonized by the Imaging-Based Coordinate System for Aging and Neurodegenerative Diseases (iSTAGING) study. The iSTAGING study comprises data acquired via various imaging protocols, scanners, data modalities, and pathologies, including more than 50,000 participants from more than 13 studies on 3 continents and encompassing a wide range of ages (22–90 years). Specifically, the current study used MRIs from the Alzheimer's Disease Neuroimaging Initiative (ADNI)[65], the UK Biobank (UKBB)[66], the Baltimore Longitudinal Study of Aging (BLSA)[67,68], the Australian Imaging, Biomarker, and Lifestyle study of aging (AIBL)[69], the Biomarkers of Cognitive Decline Among Normal Individuals in the Johns Hopkins (BIOCARD)[70], the Open Access Series of Imaging Studies (OASIS)[71], PENN, and the Wisconsin Registry for Alzheimer's Prevention (WRAP) studies[72]. In addition, whole genome sequencing (WGS) data were collected for ADNI participants; the UKBB study also consolidated the imputed genotype data (Method 5). Demographics and the number of participants from each study are detailed in Table 1. Participants provided written informed consent to the corresponding studies. The protocols of this study was approved by the University of Pennsylvania institutional review board.

## Method 3. Image processing and harmonization

A fully automated pipeline was applied to process the T1-weighted MRIs. All MRIs were first corrected for intensity inhomogeneities (ANTs: https://github.com/ANTsX/ANTs/releases/tag/v2.3.1)[73]. A multi-atlas skull stripping algorithm was applied to remove extra-cranial material (MASS: https://github.com/CBICA/MASS/releases/tag/1.1.1)[74]. Subsequently, 144 anatomical ROIs were identified in gray matter (GM, 119 ROIs), white matter (WM, 19 ROIs), and ventricles (6 ROIs) using a multi-atlas label fusion method (MUSE 3.0.5: https://github.com/CBICA/MUSE/releases/tag/3.0.5)[75]. Voxel-wise regional volumetric maps for GM and WM tissues (referred to as RAVENS)[76], were computed by spatially aligning skull-stripped images to a single subject brain template using a registration method[77]. White matter hyperintensity (WMH) volumes were calculated through a deep-learning-based segmentation method[75] built upon the U-Net architecture[78], using inhomogeneity-corrected and co-registered FLAIR and T1-weighted images. Site-specific mean and variance were estimated with an extensively validated statistical harmonization method[79] in the healthy control population and applied to the entire population while controlling for covariates.

**Table 1 | Participants and studies for the semi-synthetic and real data experiments**

| Study | N | Diagnosis | | | | Age (years) | Sex (male/%) | Semi-synthetic | Real data experiments |
|---|---|---|---|---|---|---|---|---|---|
| | | HC | MCI | AD | HTN | | | | |
| ADNI | 1533 | 472 | 784 | 277 | 0 | 73.58 ± 7.15 | 848/55.3% | 280 | 1533 |
| UKBB | 27,325 | 10,911 | 0 | 0 | 16,414 | 64.43 ± 7.48 | 13,116/48.0% | 0 | 27,325 |
| BLSA | 341 | 341 | 0 | 0 | 0 | 66.15 ± 4.84 | 146/42.8% | 341 | 0 |
| AIBL | 373 | 373 | 0 | 0 | 0 | 68.22 ± 3.99 | 147/39.4% | 373 | 0 |
| BIOCARD | 143 | 143 | 0 | 0 | 0 | 62.29 ± 5.42 | 63/44.1% | 143 | 0 |
| OASIS | 403 | 403 | 0 | 0 | 0 | 66.56 ± 5.33 | 148/36.7% | 403 | 0 |
| PENN | 107 | 107 | 0 | 0 | 0 | 67.18 ± 4.30 | 31/29.0% | 107 | 0 |
| WRAP | 90 | 90 | 0 | 0 | 0 | 63.60 ± 5.21 | 27/30.0% | 90 | 0 |

For age, the mean and the standard deviation are reported. For sex, the number of males and the percentage is presented.
*HTN* Hypertension, *HC* healthy control, *AD* clinical AD, *MCI* mild cognitive impairment, *Semi-synthetic* number of participants included in the semi-synthetic experiments, *Real data experiments* number of participants included in the real data experiments.

## Method 4. Cognitive, clinical, CSF, and plasma biomarkers

For the real data experiments, we included CSF and plasma biomarkers, *APOE ε4* alleles, and cognitive test scores provided by ADNI and UKBB. For ADNI, all measures were downloaded from the LONI website (http://adni.loni.ucla.edu). Detailed methods for CSF measurements of β-amyloid (Aβ) and phospho-tau (p-tau) are described in Hansson et al.[80] Other CSF and plasma biomarkers were measured using the multiplex xMAP Luminex platform, with details described in "Biomarkers Consortium ADNI Plasma Targeted Proteomics Project – Data Primer" (available at http://adni.loni.ucla.edu). The ADNI study has previously validated several composite cognitive scores across several domains, including ADNI-MEM[81], ADNI-EF[82], and ADNI-LAN[83]. The UKBB study provides several cognitive tests, including DSST, TMT-A, and TMT-B, etc. (https://biobank.ndph.ox.ac.uk/showcase/label.cgi?id=100026).

## Method 5. Genetic data processing and selection

Genetic analyses were performed using the whole-genome sequencing (WGS) data from ADNI and the imputed genotype data from UKBB. We performed a rigorous quality check procedure detailed in our previous papers[6-8]. which is also publicly available at our web portal: https://www.cbica.upenn.edu/bridgeport/data/pdf/BIGS_genetic_protocol.pdf.

To preselect disease-associated SNPs, we performed a manual search through the NHGRI-EBI GWAS Catalog[63]. Specifically, we used the keywords "Alzheimer's disease biomarker measurement" for AD-associated SNPs and "hypertension" for hypertension-associated SNPs. We further filtered out SNPs with reported p-values greater than or equal to $5 \times 10^{-8}$, removed SNPs with MAF smaller than 5%, included studies from European ancestries, and removed SNPs in linkage disequilibrium (i.e., window size = 2000 kb, and $r2 = 0.2$). Finally, we merged these preselected SNPs with our quality-checked WGS (AD) and imputed genotype data (hypertension). This procedure resulted in 1533 participants and 178 AD-associated SNPs for the MCI/clinical AD population and 33,541 participants and 117 hypertension-associated SNPs for the hypertension population.

## Method 6. Semi-synthetic experiments

**Data selection.** For simulated imaging features, we imposed volume decrease in predefined imaging ROIs for a proportion of (1200 out of 1739) HC participants (i.e., Pseudo-PT participants), which we referred to as the semi-synthetic datasets. For simulated genetic features, we generated fully simulated SNP data for all Pseudo-PT participants. We included 1739 participants (age < 75 and MMSE > 28) from 7 different studies (Table 1) and defined 539 participants as the real HC group and the remaining 1200 participants as the Pseudo-PT group. The Pseudo-PT group was further divided into three (M = 3) subgroups (three ground truth subtypes). Each subgroup was imposed with one specific imaging pattern and simulated genetic features – the ground truth for each subtype regarding imaging and genetic features.

**Imaging feature construction.** Three different imaging patterns were imposed on the Pseudo-PT participants within the three subgroups (Fig. 2e). The volumes of selected ROIs were randomly decreased by 0−30%, being uniformly sampled within the range. In addition, we also introduced one or two imaging-specific confounding patterns ($n_{confound} = 1$ or 2) (Fig. 2e) to one or two sets of randomly sampled Pseudo-PT participants. Importantly, these Pseudo-PT participants possessed similar confounding patterns but did not share genetic features. Details of selected ROIs for the subtypes and the confounding patterns are presented in Supplementary Data 6.

**Genetic feature construction.** We constructed an $n_{genetic}$-dimensional vector for each Pseudo-PT indicating the counts of minor alleles (0, 1, or 2) of $n_{genetic}$ SNPs. Each subtype was simulated to be associated with

four SNPs (Fig. 2d). That is, the MAF of each SNP within the subgroup was around 15% higher than the remaining participants – assuming the minor alleles are risk alleles for the simulated brain atrophy patterns. In addition, we constructed genetic-specific confounders by randomly sampling two other subgroups of Pseudo-PT and selecting four associated SNPs (i.e., confounding SNPs) for each subgroup (Fig. 2d). Selected confounding SNPs could overlap with subtype-associated SNPs in the simulation. Other non-selected SNPs had a MAF of 33% in all Pseudo-PT participants. We set $n_{genetic}$ to 100, 250, and 500 during the simulation. A higher $n_{genetic}$ indicates more complex confounding factors and leads to a more difficult task for model validations.

**Cross-validation.** On the semi-synthetic datasets with the known ground truth, we performed fifty repetitions of stratified holdout cross-validation (CV, 80% of data for training, and 20% for testing) for two purposes. First, we set different values (i.e., $5 \times 10^{-5}$, $1 \times 10^{-4}$, $2 \times 10^{-4}$, $4 \times 10^{-4}$) for gene-lr (a hyperparameter introduced in Method 1) to test the generalizability of the models. Second, we used the CV procedure to select the optimal value of gene-lr. We proposed a new metric, N-Asso-SNPs, for hyperparameter selections in the semi-synthetic and real data experiments (Method 8), which indicates the SNP-subtype associations in the test set. We calculated a log-likelihood-ratio (llr) for each SNP, as introduced in Method 8. N-Asso-SNPs equals the number of SNPs with llr>3.841. The optimal gene-lr was chosen based on the highest mean N-Asso-SNPs.

**Missing SNP test.** To test the influence of the missing SNPs in the genetic data, we randomly set 10%/20%/30% of the SNPs as missing values (NAN). We ran the Gene-SGAN model fifty times on each dataset with the optimal gene-lr selected through the abovementioned CV procedure.

**Model comparisons.** We compared Gene-SGAN with six previously proposed methods, including Smile-GAN, Deep Canonical Correlation Analysis (Deep-CCA), CCA, Multiview-spectral-clustering (MSC), Multiview-Kmeans (MKmeans), Kmeans. We ran the model fifty times on each dataset with different $n_{genetic}$ and $n_{confound}$, and derived fifty clustering accuracies. Implementation details of the six previously proposed methods can be found in Supplementary Method 3.

**The transformation function f for clinical interpretability.** We used the trained function f to *post hoc* reveal imaging patterns related to each identified subtype. For the three inputs of f, we set $z_1$ to be the one-hot vector corresponding to the subtype; additionally, we sampled 539 x and $z_2$ from the HC population and the uniform distribution $U[0,1]^{n_{z_2}}$, respectively, without replacements. The mean value of 539 calculated change ratios, $\frac{1}{539}\sum_{x,z_2}(f(x,z_1,z_2) - x)/x$, indicates the imaging patterns that drive the solution of the searched subtypes. For example, we presented the average imaging patterns characterized by 50 models trained on the dataset with $n_{genetic} = 100$ and $n_{confound} = 2$ in Fig. 2e.

**The inference function h for clinical interpretability.** The trained function h infers the genetic feature distribution corresponding to each subtype. For the two inputs of h, we set $z_1$ to be the corresponding one-hot vector and sampled 100 $z_3$ from $p(z_3) = N(0^{n_{z_3}}, 1^{n_{z_3}})$. The average of 100 outputs of h, $\frac{1}{100}\sum_{z_3} h(z_1,z_3)$, was considered the inferred MAFs of SNPs within each subtype. For example, we presented the average of MAFs inferred by 50 models trained on the dataset with $n_{genetic} = 100$ and $n_{confound} = 2$ in Fig. 2h.

## Method 7. Real data experiments

**Data selection.** For the study of MCI/AD, we included baseline participants of the ADNI study, comprised of 472 cognitively normal, 784 MCI, and 277 clinical AD participants. The 1061 MCI and clinical AD

participants were defined as the patient group, and the remaining 472 cognitively normal participants were defined as the HC group. The 144 ROIs and the 178 AD-associated SNPs were used as phenotypic and genetic features, respectively. For the study of hypertension, we selected participants from the UKBB study. Participants were defined as hypertensive patients ($N = 16,414$) if satisfying one of the three criteria: (1) systole > 130 or diastole>80; (2) history of hypertension (code 1065 or 1072 in UKBB data field: f.20002.0.0); (3) hypertension medication (UKBB data fields: f.6177.0.0 and f.6153.0.0). The HC group ($N = 10,911$) was defined as the remaining participants with systole < 130 and diastole < 80. The 144 ROIs and WMH volume (available for all UKBB participants) were used as phenotypic features, and the 117 hypertension-associated SNPs were used as genetic features. In the split-sampled experiments for hypertension, we constructed one discovery set with 10,911 HC participants and 8207 hypertensive patients and one replication set with the remaining 8207 hypertensive patients.

**Input features of Gene-SGAN.** For both studies, The ROIs/WMH volumes of all participants were first residualized to rule out the covariate (i.e., age, sex, and ICV) effects estimated in the HC group via a linear regression model. Then, the adjusted features were standardized with respect to the HC group to ensure a mean of 1 and a standard deviation of 0.1 among the HC participants for each ROI. For the genetic features, each SNP allele was recoded into 0, 1, or 2, indicating the count of minor alleles per participant. The processed imaging and genetic features were used as inputs for the Gene-SGAN model.

**Output subtypes of Gene-SGAN.** For both studies of MCI/AD and hypertension, we derived three different resolutions of clustering solutions (M = 3, 4, and 5). As introduced in Method 6, we selected gene-lr using fifty iterations of the CV procedure with N-Asso-SNPs as an evaluation metric. Specifically, the value of gene-lr ($5\times10^{-5}$, $1\times10^{-4}$, $2\times10^{-4}$, $4\times10^{-4}$) leading to the highest mean N-Asso-SNP over all three resolutions (M) was considered optimal. Next, for each M, we utilized all fifty models to derive their consensus as the final participants' subtypes (i.e., ensemble learning). In the split-sampled experiments for the hypertension population, the fifty models trained using the discovery set were applied to the 8027 patients in the replication set for deriving their subtypes.

**Method 8. Statistical analysis**
To test differences in CSF/Plasma biomarkers among the M subtypes, we performed ANOVA tests and used the Benjamin-Hochberg method to correct for multiple comparisons. Pairwise subtype comparisons were performed for other clinical (e.g., cognitive scores) and demographic variables (e.g., age, sex). For continuous variables, we utilized Mann-Whitney U tests for certain variables due to large skewness (e.g., WMH), and student's t-tests otherwise. For categorical variables (e.g., sex), the chi-squared test was used.

To test SNP-subtype associations, we performed a likelihood-ratio test on multinomial logistic regression models with subtype memberships as dependent variables. Specifically, the log-likelihood-ratio (llr) was calculated between two models fitted with and without each SNP, adjusting for covariates, including age, sex, ICV, and the first five genetic principal components. For MCI/AD, the *APOE ε4* genotype was used as another covariate. The Bonferroni method was used to correct for multiple comparisons.

**Reporting summary**
Further information on research design is available in the Nature Portfolio Reporting Summary linked to this article.

## Data availability

Data used for this study were provided from several individual studies via data sharing agreements that did not include permission for us to further share the data. However, data from ADNI are available from the ADNI database (adni.loni.usc.edu) upon registration and compliance with the data usage agreement. Data from the UKBB are available to registered researchers upon request from the UKBB website (https://www.ukbiobank.ac.uk/). Data from the BLSA study are available upon request at https://www.blsa.nih.gov/how-apply. Data from the AIBL study are available upon request at https://aibl.org.au/. Data from the OASIS study are available upon request at https://www.oasis-brains.org/. The detailed requirements for requesting and accessing each dataset are listed on the corresponding website. Data requests for Biocard, Penn, and WRAP datasets should be directed to M.A. (malbert9@jhmi.edu), D.W. (david.wolk@pennmedicine.upenn.edu), and S.J. (scj@medicine.wisc.edu), respectively, who will provide requirements and restrictions on data-use via data-use agreements. Participant-level derived subtypes generated in this study will be provided within one month of receiving approval granted from respective studies. The GWAS summary statistics generated in this study are provided in Supplementary Data files. All data supporting the findings described in this manuscript are available in the article and its Supplementary Information files, and from the corresponding author upon request.

## Code availability

The software Gene-SGAN is available as a published PyPI package. Detailed information about software installation, usage, and license can be found at: https://pypi.org/project/GeneSGAN/. Custom code can be found at: https://github.com/zhijian-yang/GeneSGAN[84], which is archived in Zenodo with the identifier [https://doi.org/10.5281/zenodo.10058768].

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

## Acknowledgements

Data used in this study are part of the iSTAGING study (representative: Christos Davatzikos), the Preclinical AD Consortium (M.S.A.), the Alzheimer's Disease Neuroimaging Initiative (Michael W. Weiner), the Baltimore Longitudinal Study of Aging (S.M.R.), and the AI4AD Consortium (Christos Davatzikos, Paul M. Thompson, Andrew J. Saykin, Li Shen, and Heng Huang). The iSTAGING consortium is a multi-institutional effort funded by NIA by RF1 AG054409. The AI4AD was funded by NIH under U01AG068057. The Baltimore Longitudinal Study of Aging neuroimaging study is funded by the Intramural Research Program, National Institute on Aging, National Institutes of Health and by HHSN271201600059C. The BIOCARD study is in part supported by NIH grant U19-AG033655. Other supporting grants include 5U01AG068057-02, 1U24AG074855-01. S.R.H and J.C.M are supported by multiple grants and contracts from NIH. A.A. was supported by grants 191026 and 206795 from the Swiss National Science Foundation. This research has been conducted using the UK Biobank Resource under Application Number 35148. Data used in the preparation of this article were in part obtained from the Alzheimer's Disease Neuroimaging Initiative (ADNI) database (adni.loni.usc.edu). As such, the investigators within the ADNI contributed to the design and implementation of ADNI and/or provided data but did not participate in the analysis or writing of this report. A complete listing of ADNI investigators can be found at: http://adni.loni.usc.edu/wpcontent/uploads/how_to_apply/ADNI_Acknowledgement_List.pdf. ADNI is funded by the National Institute on Aging, the National Institute of Biomedical Imaging and Bioengineering, and through generous contributions from the following: AbbVie, Alzheimer's Association; Alzheimer's Drug Discovery Foundation; Araclon Biotech; BioClinica, Inc.; Biogen; Bristol-Myers Squibb Company; CereSpir, Inc.; Cogstate; Eisai Inc.; Elan Pharmaceuticals, Inc.; Eli Lilly and Company; EuroImmun; F. Hoffmann-La Roche Ltd and its affiliated company Genentech, Inc.; Fujirebio; GE Healthcare; IXICO Ltd.; Janssen Alzheimer Immunotherapy Research & Development, LLC.; Johnson & Johnson Pharmaceutical Research & Development LLC.; Lumosity; Lundbeck; Merck & Co., Inc.; Meso Scale Diagnostics, LLC.; NeuroRx Research; Neurotrack Technologies; Novartis Pharmaceuticals Corporation; Pfizer Inc.; Piramal Imaging; Servier; Takeda Pharmaceutical Company; and Transition Therapeutics. The Canadian Institutes of Health Research is providing funds to support ADNI clinical sites in Canada. Private sector contributions are facilitated by the Foundation for the National Institutes of Health (www.fnih.org). The grantee organization is the Northern California Institute for Research and Education, and the study is coordinated by the Alzheimer's Therapeutic Research Institute at the

University of Southern California. ADNI data are disseminated by the Laboratory for Neuro Imaging at the University of Southern California. Mr. Yang had full access to all the data in the study. He took responsibility for the integrity of the data and the accuracy of data analysis.

## Author contributions

Study concept and design: Z.Y. and C.D. Statistical analysis: Z.Y. Data interpretation: Z.Y., J.W., H.S., I.M,N. and C.D. Data collection and processing: Z.Y., J.W., Y.C., G.E., A.A., E.M., R.M., D.S., S.T.G., J.C., C.L.M., P.M., J.F., L.F., M.S.A., S.C.J., J.C.M., P.L., D.S.M., T.L.S.B., D.A.W., L.S., S.R., I.M.N., C.D. Manuscript critical revision and submission approval: Z.Y., J.W., A.A., Y.C., G.E., E.M., R.M., D.S., S.T.G., J.C., M.H., C.L.M P.M., J.F., L.F., M.S.A., S.J., J.M., P.L., D.M., T.B., D.W., L.S., J.B., H.S., S.R., I.M.N, C.D.

## Competing interests

The authors declare no competing interests

## Additional information

Zhijian Yang [1,2], Junhao Wen [1,3], Ahmed Abdulkadir [4], Yuhan Cui[1], Guray Erus [1], Elizabeth Mamourian [1], Randa Melhem[1], Dhivya Srinivasan[1], Sindhuja T. Govindarajan[1], Jiong Chen[1], Mohamad Habes[5], Colin L. Masters[6], Paul Maruff[6], Jurgen Fripp[7], Luigi Ferrucci [8], Marilyn S. Albert[9], Sterling C. Johnson [10], John C. Morris[11], Pamela LaMontagne [12], Daniel S. Marcus[12], Tammie L. S. Benzinger [11,12], David A. Wolk[13], Li Shen [14], Jingxuan Bao [14], Susan M. Resnick[15], Haochang Shou [14], Ilya M. Nasrallah [1,16] & Christos Davatzikos [1] ✉

[1]Artificial Intelligence in Biomedical Imaging Laboratory (AIBIL), Center for and Data Science for Integrated Diagnostics (AI2D), Perelman School of Medicine, University of Pennsylvania, Philadelphia, PA, USA. [2]Graduate Group in Applied Mathematics and Computational Science, University of Pennsylvania, Philadelphia, PA, USA. [3]Laboratory of AI and Biomedical Science (LABS), Stevens Neuroimaging and Informatics Institute, Keck School of Medicine of USC, University of Southern California, Los Angeles, CA, USA. [4]Laboratory for Research in Neuroimaging, Department of Clinical Neurosciences, Lausanne University Hospital (CHUV) and University of Lausanne, Lausanne, Switzerland. [5]Biggs Alzheimer's Institute, University of Texas San Antonio Health Science Center, San Antonio, TX, USA. [6]The Florey Institute of Neuroscience and Mental Health, The University of Melbourne, Parkville, VIC, Australia. [7]CSIRO Health and Biosecurity, Australian e-Health Research Centre CSIRO, Brisbane, QLD, Australia. [8]Translational Gerontology Branch, Longitudinal Studies Section, National Institute on Aging, National Institutes of Health, MedStar Harbor Hospital, 3001 S. Hanover Street, Baltimore, MD, USA. [9]Department of Neurology, Johns Hopkins University School of Medicine, Baltimore, MD, USA. [10]Wisconsin Alzheimer's Institute, University of Wisconsin School of Medicine and Public Health, Madison, WI, USA. [11]Knight Alzheimer Disease Research Center, Washington University in St. Louis, St. Louis, MO, USA. [12]Mallinckrodt Institute of Radiology, Washington University School of Medicine, St. Louis, MO, USA. [13]Department of Neurology, University of Pennsylvania, Philadelphia, PA, USA. [14]Department of Biostatistics, Epidemiology and Informatics, University of Pennsylvania, Philadelphia, PA, USA. [15]Laboratory of Behavioral Neuroscience, National Institute on Aging, Baltimore, MD, USA. [16]Department of Radiology, University of Pennsylvania, Philadelphia, PA, USA. ✉e-mail: Christos.Davatzikos@pennmedicine.upenn.edu

