## [Peer Review File · Nature Communications]

Gene-SGAN: discovering disease subtypes with imaging and genetic signatures via multi-view weakly-supervised deep clusteringREVIEWER COMMENTS

Reviewer #1 (Remarks to the Author):

The authors proposed a multi-view, weakly-supervised deep clustering method (Gene-SGAN) by combining generative adversarial network and variational inference. The Gene-SGAN is well-designed, which can identify disease-related subtypes by a generative adversarial network and link to genetic variants by variational inference. The authors apply Gene-SGAN to identify four AD subtypes from the ADNI dataset (1533 participants) and five hypertension subtypes from the UKBB study (16,414 participants), which were also associated with the genetic variants. This is a very interesting study, which may have potentially important implications in subtype identification associated with specific disease-related features (e.g. genetic features). However, there are also some concerns.

1. The authors validate the generalizability, interpretability, and robustness of Gene-SGAN in semi-synthetic experiments through cross-validation. However, the cross-validation was only performed on the target group whereas the same reference group was used in all validation analysis, which may cause higher accuracy. Different reference groups may have an impact on the low-dimensional latent space. For example, in the AD subtype analysis, when the reference group was replaced with the healthy controls in the UKBB, could the same abnormal neuroanatomical and genetic patterns of AD subtypes be replicated? The authors should perform an analysis of the impact of the reference group selection on subtype identification, especially when applying the Gene-SGAN to an independent study in the future.

2. In the AD and hypertension subtype analysis, the authors set the number of subtypes as 3, 4 and 5, and present the results with four AD subtypes and five hypertension subtypes according to previous studies. Could the authors provide an approach for determining the optimal number of clusters for the Gene-SGAN? In addition, when the number of hypertension subtypes increased from 3 to 4, the emergence of a new subtype mainly results from the combination of three subtypes (4M-H2: 1173 3M-H1 + 1938 3M-H2 + 699 3M-H3), rather than further subdivision of a specific subtype. Could the authors give more explanation about it?

3. The authors identified four AD subtypes with distinct imaging signatures, which seem like subtypes of different disease processes. Previous studies also identified parieto-occipital or subcortical dominant AD subtype based on the same dataset (Ten Kate M, Dicks E, Visser P J, et al. Atrophy subtypes in prodromal Alzheimer's disease are associated with cognitive decline[J]. *Brain*, 2018; Zhang X, Mormino E C, Sun N, et al. Bayesian model reveals latent atrophy factors with dissociable cognitive trajectories in Alzheimer's disease[J]. *Proceedings of the National Academy of Sciences*, 2016.). Could the authors give more explanation about the differences in the identified subtypes?

4. In the hypertension subtype analysis, the authors split the patient data into the discovery set (N=8207) and the replication set (N=8207) to test the replicability rate of the SNP-subtype associations. The replication rate of the abnormal neuroanatomical patterns should also be displayed. If possible, the healthy controls should also be independent in the discovery and replication sets to test the replicability of the identified subtypes.

5. The authors introduced one or two imaging-specific confounding patterns to one or two sets of randomly sampled Pseudo-PT participants that did not share genetic features. Could the imaging-specific confounding patterns be identified by the latent variable z_2 , which can capture the phenotype-specific variations?

6. The equation misses a parenthesis in line 893.

Reviewer #2 (Remarks to the Author):

Summary:

The paper presents a new framework to analyze disease heterogeneity using neuroimaging (structural MRI) and genetic data. The work aims to achieve this by a semi-supervised clustering approach using a generative adversarial network (GAN) and variational inference (VI) models to extract phenotypic and genetic information, respectively. Together, these models infer three latent variables encoding phenotype as well as genetic, exclusively phenotypic and exclusively genetic variations that define the mapping from reference (control) phenotypic features to various sets of target phenotypic features in the patient population. The framework is tested on two real and also semi-synthetic datasets to identify disease subtypes and the associated phenotypic and genetic features.

This work builds upon previous frameworks that do not inculcate genetic information into account. Adding genetic information into the framework and also separating the overlapping and non-overlapping components with the phenotypic information is an important addition.

The overall approach is promising, as it is able to compute subtypes and characterize the associated patterns by bucketing them into a mix of genetic and phenotypic variables.

Comments:

-The discovered patterns look in many cases to be highly similar among subtypes, varying along a mostly global scale (or to be negative vs positive versions of a very similar pattern). It would be helpful to have the authors assess how similar the patterns are to one another, and also show differences between them, as this is important for interpretation (to what degree are they separable, etc.)

-In the methods section, the authors should clarify in further detail about why genetic features are used as target variables and not as reference variables.

-While replication across datasets is done, there could be more analysis to show whether the obtained subtypes and corresponding phenotypic and genetic signatures (encoded by z_1 , z_2 , z_3) are stable across cross-validation repetitions on the same datasets, as done for testing accuracy.

-Similarly, there could be more analysis to show if there is stability for different model orders (number of subtypes) in the imaging and genetic patterns associated with each subtype. The authors could discuss whether the model being used it is expected to output different/unstable associated patterns learned in variables z_1 , z_2 , z_3 for the obtained subtypes under different model orders.

-For robust interpretability, the 'top' subtypes obtained for different model orders and different cross-validation sets should be stable in terms of their corresponding encoded patterns.

-The authors may want to refrain from using terminologies implying causality (like "-driven") as the approach is a semi-supervised clustering approach learning associative patterns for the computed subtypes.

Dear Editor and Reviewers:

We are grateful to the reviewers for their constructive comments, which have been very helpful in revising our manuscript and in improving its quality.

We addressed the questions raised by the reviewers as follows. Changes and corrections are marked in **red** in the revised manuscript.

Reviewer #1 comments and responses:

1. The authors validate the generalizability, interpretability, and robustness of Gene-SGAN in semi-synthetic experiments through cross-validation. However, the cross-validation was only performed on the target group whereas the same reference group was used in all validation analysis, which may cause higher accuracy. Different reference groups may have an impact on the low-dimensional latent space. For example, in the AD subtype analysis, when the reference group was replaced with the healthy controls in the UKBB, could the same abnormal neuroanatomical and genetic patterns of AD subtypes be replicated? The authors should perform an analysis of the impact of the reference group selection on subtype identification, especially when applying the Gene-SGAN to an independent study in the future.

Response: We would like to thank the reviewer for this valuable comment. We agree that it is important to evaluate the robustness of the model's performance in relation to independent reference groups. We would like to first clarify that, in our semi-synthetic experiments, we did perform stratified cross-validations with each fold using 80% of the target and 80% of the reference group as training data. The trained models could be directly applied to the target group in the test set to derive their latent variables, without requiring subject-level reference data. Prior to this, only the parameters estimated from the training reference data were applied to standardize test PT data. We have now provided a clearer description of this process at the end of **Method 1**.

Beyond that, we have also conducted more concrete analyses, as suggested by the reviewer, regarding the impact of reference groups. First, in our cross-validation experiments on the semi-synthetic datasets, we additionally calculated clustering accuracies by applying the trained models to test PT data that were standardized with respect to 20% test reference data (**Supplementary eFigure 1**). Second, in our experiments on the MCI/AD dataset, we tested the replicability of subtypes under two different scenarios: 1) independent reference in model application, where we applied the trained models to MCI/AD participants standardized with respect to the HC group from UKBB; and 2) different reference in model training, where we retrained the models using MCI/AD participants from ADNI and HC participants from UKBB (**Supplementary eResult 1.2**). Finally, in split-sampled experiments on the hypertension dataset, we additionally half-divided the HC group, and similarly analyzed the reproducibility of subtypes with an independent reference group in model application or model training (**Supplementary eResult 1.1** and **Supplementary eTable 5**).

2. In the AD and hypertension subtype analysis, the authors set the number of subtypes as 3, 4 and 5, and present the results with four AD subtypes and five hypertension subtypes according to previous studies. Could the authors provide an approach for determining the optimal number of clusters for the Gene-SGAN?

Response: We agree that an approach for determining the optimal number of clusters is important for clustering algorithms. In the existing literature, researchers have used agreements among cross-validated or repetitively trained models to derive the ideal number of clusters^{1,2}. Metrics such as adjusted random index (ARI)³ and adjusted mutual information (AMI)¹ have been employed to measure these agreements. However, in our experiments with the Gene-SGAN model, these metrics tended to favor smaller numbers of clusters rather than the correct one. We have introduced the *N-Asso-SNPs* metric for selecting hyperparameters when fixing the number of clusters. However, we found that this metric consistently increased with the number of clusters without adjustments. Therefore, we proposed an adjusted *N-Asso-SNPs* metric, which, in most cases, could suggest the ground truth number of clusters (M=3) in the semi-synthetic experiments. For further details on the adjustments and experimental results, please refer to **Supplementary eMethod 5** and **Supplementary eFigure 1**.

3. In addition, when the number of hypertension subtypes increased from 3 to 4, the emergence of a new subtype mainly results from the combination of three subtypes (4M-H2: 1173 3M-H1 + 1938 3M-H2 + 699 3M-H3), rather than further subdivision of a specific subtype. Could the authors give more explanation about it?

Response: In our current model framework, participants in the additional subtype were assigned to the nearest subtype with less confidence when the number of clusters was decreased. However, we could still observe consistent subtyping and latent variables across different scales (number of clusters) (**Supplementary eFigure 2 and 3**). We agree that a hierarchical approach, subdividing a specific subtype, might be more advantageous in certain scenarios. Therefore, we have included this potential improvement in the discussion section. However, we currently don't have much biological evidence that the type of clustering we investigate would benefit from hierarchical constraints.

4. The authors identified four AD subtypes with distinct imaging signatures, which seem like subtypes of different disease processes. Previous studies also identified parieto-occipital or subcortical dominant AD subtype based on the same dataset (Ten Kate M, Dicks E, Visser P J, et al. Atrophy subtypes in prodromal Alzheimer's disease are associated with cognitive decline[J]. *Brain*, 2018; Zhang X, Mormino E C, Sun N, et al. Bayesian model reveals latent atrophy factors with dissociable cognitive trajectories in Alzheimer's disease[J]. *Proceedings of the National Academy of Sciences*, 2016.). Could the authors give more explanation about the differences in the identified subtypes?

Response: Based on the suggestions from the reviewer, we have included one paragraph of discussion on the relationship between our findings and results obtained from other subtyping studies. Also, as emphasized in the first paragraph of the Discussion section, Gene-SGAN uses the generative model that maps measurements from the reference population to the target population, thereby aiming to capture variations of disease effects, rather than variations of individuals' phenotypes, which might be influenced by a number of disease-unrelated factors.

The discussion paragraph is copied here:

The identified AD-related subtypes exhibit similarities to other neuroimaging-based clustering studies, including the identification of MTL- and cortical-predominant patterns⁴⁻⁷. However,

Gene-SGAN's subtypes were specifically refined to maximize genetic associations. As such, it focuses on deriving imaging endophenotypes, rather than just phenotypic clusters. For instance, we observed two extreme subtypes characterized by highly focal hippocampal atrophy and preserved brain volume, showing significant differences in rs429358 (*APOE*) with the highest and lowest EAFs, respectively. The minimal differences in *APOE* $\epsilon 2$ and *APOE* $\epsilon 4$ among the other neuroimaging-based subtypes⁵⁻⁷ indirectly verify the refinements provided by Gene-SGAN. Moreover, in direct comparisons to the subtypes presented in Yang et al.⁴, Gene-SGAN's subtypes demonstrate much stronger genetic associations, further validating the effectiveness of refinements. It is worth noting that certain previously identified atrophy subtypes, such as occipital atrophy patterns⁷, are primarily captured by latent variables z_2 , suggesting their limited genetic associations.

5. In the hypertension subtype analysis, the authors split the patient data into the discovery set (N=8207) and the replication set (N=8207) to test the replicability rate of the SNP-subtype associations. The replication rate of the abnormal neuroanatomical patterns should also be displayed. If possible, the healthy controls should also be independent in the discovery and replication sets to test the replicability of the identified subtypes.

Response: We concur that analyzing the replication of neuroanatomical patterns would provide more valuable insights. **Supplementary eFigure 6** presents the imaging patterns associated with the five subtypes in both discovery and replication sets, revealing a high level of consistency between the two sets for each subtype. Furthermore, as addressed in our responses to question 1, we have conducted comprehensive analyses to assess the replicability of subtypes using independent healthy control groups, both during model application and in the training processes. In both scenarios, we observed a strong replicability of the identified subtypes with independent reference groups.

6. The authors introduced one or two imaging-specific confounding patterns to one or two sets of randomly sampled Pseudo-PT participants that did not share genetic features. Could the imaging-specific confounding patterns be identified by the latent variable z_2 , which can capture the phenotype-specific variations?

Response: We appreciate the reviewer's suggestion. Evaluating the imaging patterns represented by the latent variables z_2 would indeed enhance our understanding of the model's performance. In **Supplementary eFigure 1**, we have revealed the ROIs associated with the first two z_2 PCs of models trained on the semi-synthetic dataset, which incorporates two imaging-specific confounding patterns. Our findings clearly demonstrate that the first two PCs predominantly captured the simulated imaging-specific variations, specifically atrophy in subcortical and occipital regions.

7. The equation misses a parenthesis in line 839.

Response: We appreciate the reviewer for catching this mistake. We have corrected it in the manuscript now.

Reviewer #2 comments and responses:

1. The discovered patterns look in many cases to be highly similar among subtypes, varying along a mostly global scale (or to be negative vs positive versions of a very similar pattern). It would be helpful to have the authors assess how similar the patterns are to one another, and also show differences between them, as this is important for interpretation (to what degree are they separable, etc.)

Response: We appreciate the comment, and we assume that the reviewer is referring to the AD-related subtypes, which seem to display this characteristic, rather than the hypertension-related subtypes, which are quite distinct. We would like to first admit that our original choices of color scale make some regional differences less clear. Thus, we have updated the color scale of **Figure 3** and **Supplementary eFigure 2** to enhance the clarity of the information presented. Additionally, to resolve challenges in distinguishing certain AD-related subtypes, we performed subtype vs subtype comparisons among the last three subtypes of AD beyond subtype vs HC group comparisons, allowing us to directly measure and visualize their differences. Through these comparisons, we could more clearly observe that 4M-A2 exhibits focal hippocampal atrophy without significant atrophy in other brain regions, while 4M-A4 predominantly displays cortical atrophy with very sparing MTL atrophy. 4M-A1 has preserved brain volumes, thus showing very significant differences from other subtypes. Although it could be perceived as a negative version of other subtypes, it remains essential to identify this particular subtype due to its association with corresponding genetic protective factors, as elaborated in the discussion section. Finally, we would like to point out that, at a high level, we don't seek to find subtypes with non-overlapping patterns. This is because brain aging and neurodegenerative diseases are affected by a range of underlying neuropathological processes that partially overlap and co-occur. Furthermore, these processes also impact overlapping regions of the brain. As a result, it is most likely that seeking out entirely separate patterns of brain atrophy might not be appropriate in our application.

2. In the methods section, the authors should clarify in further detail about why genetic features are used as target variables and not as reference variables.

Response: We agree that providing further clarification on this aspect would better help readers understand the model framework. Based on the reviewer's suggestions, we have included additional explanations in the method section.

The explanation paragraph is copied here:

Different from the **Phenotype step**, we do not include the reference group for learning a transformation of genetic features due to their innateness and immutability. Also, we opt not to incorporate a feature selection mechanism with respect to reference data within the model framework, as it requires a large dataset to comprehensively identify disease-associated SNPs. Instead, we pre-select candidate SNPs associated with the disease of interest using the GWAS-Catalog⁸ online portal. The **Gene step** encourages the clustering solution of the target group to be associated with the candidate genetic features. This approach not only provides more comprehensive selection of candidate SNPs but also makes Gene-SGAN more available to users lacking large imaging-genomic datasets.

3. While replication across datasets is done, there could be more analysis to show whether the obtained subtypes and corresponding phenotypic and genetic signatures (encoded by z_1 , z_2 , z_3) are stable across cross-validation repetitions on the same datasets, as done for testing accuracy.

Response: We agree with the reviewer that it is important to derive stable subtypes to guarantee the reproducibility of model results. Recognizing the significance of model consensus in deriving robust subtypes, we employed a 5-fold nested cross-validation (CV) to assess the replicability of obtained subtypes and corresponding ancillary latent variables z_2 and z_3 . Specifically, we divided both PT and HC data into five folds. In each iteration of the outer loop, we used 4 folds of data (80%), performed a 50-repetition 20% holdout CV as the inner loop, and derived z_1 , z_2 , z_3 through the consensus of 50 models. We evaluated the replicability of z_1 , z_2 , z_3 among 5 consensus results obtained from the outer loop. Adjusted Mutual Information (AMI) was used for measuring the replicability of the derived subtypes (encoded by z_1). The correlations of the first three PCs were used for measuring the replicability of z_2 and z_3 . Detailed results could be found in **Supplementary eResult 1, eTable 4, and eTable 6**. We could observe notable replicabilities considering the relative independence of training data as well as the randomness of the model initializations and training processes. **Supplementary eResult 1.1** provides further evidence that even with completely separate training sets, the Gene-SGAN model could still achieve replicable subtypes.

4. Similarly, there could be more analysis to show if there is stability for different model orders (number of subtypes) in the imaging and genetic patterns associated with each subtype. The authors could discuss whether the model being used is expected to output different/unstable associated patterns learned in variables z_1 , z_2 , z_3 for the obtained subtypes under different model orders. For robust interpretability, the ‘top’ subtypes obtained for different model orders and different cross-validation sets should be stable in terms of their corresponding encoded patterns.

Response: We agree with the reviewer that it is important to derive stable subtypes for different model scales (number of subtypes). First, we would like to highlight that, from **Supplementary eFigure 2 and 3**, we could observe that the Gene-SGAN model did identify consistent and refined imaging subtypes from a coarse to refined resolution with an increasing number of clusters. Additionally, based on the reviewer’s suggestion, we have performed more comprehensive experiments to examine the replicability of subtypes derived with different model scales. As in our response to question 3, we employed AMI to measure the replicability of subtypes (encoded by z_1), and we examined the associations of PCs for z_2 and z_3 . Detailed results can be found in **Supplementary eTable 7 and 8**. We have also included more discussions on the reproducibility of results in **Supplementary eResult 1.3**.

5. The authors may want to refrain from using terminologies implying causality (like “-driven”) as the approach is a semi-supervised clustering approach learning associative patterns for the computed subtypes.

Response: We appreciate the reviewer for pointing out this problem, and we agree that the original terminology might be misleading. We have changed all “-driven” to be “-associated” now.

- 1 Vinh, N., Epps, J. & Bailey, J. Information Theoretic Measures for Clusterings Comparison: Variants, Properties, Normalization and Correction for Chance. *Journal of Machine Learning Research* **11**, 2837-2854 (2010).
- 2 Vinh, N., Epps, J. & Bailey, J. *Information theoretic measures for clusterings comparison: Is a correction for chance necessary?* , (2009).
- 3 Hubert, L. & Arabie, P. Comparing partitions. *Journal of Classification* **2**, 193-218 (1985). <https://doi.org:10.1007/BF01908075>
- 4 Yang, Z. *et al.* A deep learning framework identifies dimensional representations of Alzheimer's Disease from brain structure. *Nature Communications* **12** (2021). <https://doi.org:10.1038/s41467-021-26703-z>
- 5 Dong, A. *et al.* Heterogeneity of neuroanatomical patterns in prodromal Alzheimer's disease: links to cognition, progression and biomarkers. *Brain* **140**, 735-747 (2017). <https://doi.org:10.1093/brain/aww319>
- 6 Zhang, X. *et al.* Bayesian model reveals latent atrophy factors with dissociable cognitive trajectories in Alzheimer's disease. *Proc Natl Acad Sci U S A* **113**, E6535-e6544 (2016). <https://doi.org:10.1073/pnas.1611073113>
- 7 Ten Kate, M. *et al.* Atrophy subtypes in prodromal Alzheimer's disease are associated with cognitive decline. *Brain* **141**, 3443-3456 (2018). <https://doi.org:10.1093/brain/awy264>
- 8 Buniello, A. *et al.* The NHGRI-EBI GWAS Catalog of published genome-wide association studies, targeted arrays and summary statistics 2019. *Nucleic Acids Res* **47**, D1005-d1012 (2019). <https://doi.org:10.1093/nar/gky1120>

REVIEWERS' COMMENTS

Reviewer #1 (Remarks to the Author):

The authors have addressed most of the replicability problems and cross-validation results in the supplementary file, however, considering the importance of the reliability of the imaging and genetic biomarkers, the corresponding supplementary figures such as eFig5 and eFig6 should be included in the main text, or the authors should find a way to add some subplots.

Moreover, the approach is a semi-supervised clustering approach learning associative patterns for the computed subtypes related to genetic factors. Therefore, the potential use and biological plausibility of the identified subtypes in either genetic or imaging phenotypic applications should be clarified more in the discussion.

Reviewer #3 (Remarks to the Author):

I have carefully reviewed the authors' responses, specifically to the concerns raised by Reviewer #2, to assess the changes they made in light of those concerns. Based on my review, the authors have addressed Reviewer #2's comments and provided clarifications on pattern differentiation, methodology, and model stability.

Reviewer #2's first concern was that the discovered patterns appeared very similar across subtypes, seemingly varying mainly at a global scale or looking like positive or negative versions of a very similar pattern. The reviewer requested clarification on how similar these patterns were to each other and wanted the authors to highlight the distinctions.

In their response, the authors updated the color scale to modify the clarity of the information in Figure 3 and Supplementary eFigure 2. They also mentioned additional comparisons made between subtypes to address the differentiation issue related to AD subtypes. According to eFigure 2, the authors analyzed differences between the HC group (i.e., participants with normal cognitive function) and each AD-related subtype through voxel-wise group comparisons. This comparison was done referencing $xM-A_y$, which represents the y -th AD-related subtype derived using Gene-SGAN with cluster number $M=x$. The authors' reply aligns with their figures and explanations.

Reviewer #2's second question sought an explanation in the methods section on why genetic features were used as target variables and not as reference variables.

The authors provided a rationale and detailed their specific approach within the model framework. Their explanation clarifies their methodology and the reasoning behind using genetic features as target variables.

In Reviewer #2's third question, the reviewer highlighted the need for a more detailed analysis of the stability of latent variables and subtypes in the original dataset through cross-validation iterations.

In response, the authors employed a 5-fold nested cross-validation, dividing the PT and HC data into five subsets. In each outer loop, four subsets (80% of the data) were used to carry out 50 rounds of 20% holdout cross-validation in the inner loop. A consensus from the 50 models was then used to derive z1, z2, and z3. They used AMI to measure the replicability of the subtypes, and correlations of the first three principal components to assess the replicability of z2 and z3. The findings show replicability considering model initialization, randomness in the learning process, and the relative independence of the training data.

Reviewer #2's fourth question highlighted the following points: The stability of imaging and genetic patterns related to each subtype across various model orders (number of subtypes) should be examined, and for robust interpretation, the main subtypes obtained across different model orders and cross-validation sets should remain stable in terms of their encoding patterns.

The authors showed the consistency of subtypes and latent variables across different model sizes in eTable 7 & 8. Results in eFigure 2 & 3 indicate that the primary subtypes do not change substantially as the cluster number or model order increases.

Reviewer #2's fifth concern was related to the use of terms implying causality that didn't align with the study's methodology. The authors acknowledged this issue and modified the terminology as recommended by the reviewer.

After reviewing the authors' responses to Reviewer #2, the authors addressed the reviewer's questions. I did not identify any areas that would require further verification.

Dear Editor and Reviewers:

We are grateful to the reviewers for their constructive comments, which have been very helpful in revising our manuscript and in improving its quality.

We addressed the questions raised by the reviewers as follows. Changes and corrections are marked in **red** in the revised manuscript.

Reviewer #1 comments and responses:

1. The authors have addressed most of the replicability problems and cross-validation results in the supplementary file, however, considering the importance of the reliability of the imaging and genetic biomarkers, the corresponding supplementary figures such as eFig5 and eFig6 should be included in the main text, or the authors should find a way to add some subplots.

Response: We would like to thank the reviewer for this suggestion. We agree that eFig5 and eFig6 are important for readers to better understand differences among subtypes as well as the reproducibility of imaging signatures corresponding to different subtypes. We have now included Supplementary eFig5 as part of Figure 3 in the main manuscript. However, due to the large space requirement of eFig6, we have kept it in the supplementary material. Nevertheless, in the main manuscript, we have included a brief paragraph emphasizing the replicability and reliability of the imaging patterns derived using Gene-SGAN, with a reference to eFig6 for further details.

The paragraph is copied here:

We further performed split-sampled analyses (**Method 7**) to test the replicability of imaging patterns associated with five subtypes. Patient data was evenly divided into the discovery set (N=8207) and the replication set (N=8207). The Gene-SGAN model, retrained on the discovery set, was applied to both sets to re-derive the five hypertension-related subtypes. Highly consistent imaging signatures were observed between discovery and replication sets (**Supplementary eFigure 5**), which also have strong agreements with the subtypes derived from the entire dataset (**Figure 4a**).

2. The approach is a semi-supervised clustering approach learning associative patterns for the computed subtypes related to genetic factors. Therefore, the potential use and biological plausibility of the identified subtypes in either genetic or imaging phenotypic applications should be clarified more in the discussion.

Response: We agree that more clarifications regarding the applications would enhance comprehension of our approach. Based on the suggestion, we have enriched the paragraph that discusses plausible applications of the identified subtypes, specifically in precision diagnostics, clinical trials, disease research, and drug discovery.

The paragraph is copied here:

The potential clinical impact of Gene-SGAN is versatile. In general, it helps dissect the heterogeneity of diseases into relatively more neuroanatomical subtypes that also have genetic underpinnings, and hence it contributes to precision diagnostics that can have downstream effects

on any subsequent analysis. For example, deriving robust disease-related subtypes may help improve classification performance for individualized disease diagnosis and prognosis. Moreover, modeling disease heterogeneity provides new patient stratification and treatment evaluation tools for future clinical trials, which remain important in the setting of mixed results and clinical limitations of anti-amyloid treatments. It is well recognized that evaluating treatment responses within relatively more homogeneous subgroups of patients can significantly increase the power of clinical trials. Our results also suggest that disease subtyping via Gene-SGAN could augment our ability to detect significant imaging and genomic characteristics of AD, which would be diluted in case-control comparisons due to the underlying heterogeneity. Finally, Gene-SGAN subtypes are genetically relevant by modeling, which serves as a reliable endophenotype to pinpoint potential causal genetic variants for drug repurposing and discovery.

Reviewer #3 comments and responses:

1. After reviewing the authors' responses to Reviewer #2, the authors addressed the reviewer's questions. I did not identify any areas that would require further verification.

Response: We would like to thank the reviewer for the careful review and positive feedback on our responses to Reviewer #2.